# High-throughput and high-efficiency sample preparation for single-cell proteomics using a nested nanowell chip

Jongmin Woo [1], Sarah M. Williams[1], Lye Meng Markillie [1], Song Feng [2], Chia-Feng Tsai[2], Victor Aguilera-Vazquez[1], Ryan L. Sontag[2], Ronald J. Moore [2], Dehong Hu [1], Hardeep S. Mehta[1], Joshua Cantlon-Bruce[3,4], Tao Liu [2], Joshua N. Adkins [2], Richard D. Smith [2], Geremy C. Clair [2], Ljiljana Pasa-Tolic [1] & Ying Zhu [1✉]

Global quantification of protein abundances in single cells could provide direct information on cellular phenotypes and complement transcriptomics measurements. However, single-cell proteomics is still immature and confronts many technical challenges. Herein we describe a nested nanoPOTS (N2) chip to improve protein recovery, operation robustness, and processing throughput for isobaric-labeling-based scProteomics workflow. The N2 chip reduces reaction volume to <30 nL and increases capacity to >240 single cells on a single microchip. The tandem mass tag (TMT) pooling step is simplified by adding a microliter droplet on the nested nanowells to combine labeled single-cell samples. In the analysis of ~100 individual cells from three different cell lines, we demonstrate that the N2 chip-based scProteomics platform can robustly quantify ~1500 proteins and reveal membrane protein markers. Our analyses also reveal low protein abundance variations, suggesting the single-cell proteome profiles are highly stable for the cells cultured under identical conditions.

[1] Environmental Molecular Sciences Laboratory, Pacific Northwest National Laboratory, Richland, WA 99354, USA. [2] Biological Sciences Division, Pacific Northwest National Laboratory, Richland, WA 99354, USA. [3] Scienion AG, Volmerstraße 7, 12489 Berlin, Germany. [4] Cellenion SASU, 60 Avenue Rockefeller, Bâtiment BioSerra2, 69008 Lyon, France. ✉email: ying.zhu@pnnl.gov

With the success of single-cell genomics and transcriptomics, there is a growing demand for high-throughput single-cell proteomics (scProteomics) technologies. Global profiling of protein expression in individual cells can identify protein markers specific to certain cell populations in heterogeneous mixtures, provide molecular evidence of cellular function phenotypes, and help to identify critical post-translational modifications that regulate protein activities[1–3]. Despite its transformative potential, scProteomics still lags behind single-cell transcriptomics in terms of coverage, measurement throughput, and quantitation accuracy[4].

Most reported mass-spectrometry-based scProteomics technologies can be classified based upon whether they make use of isotopic labeling; i.e., they are either label-free or use isobaric labeling. In the label-free methods[5–10], single cells are individually processed and analyzed with liquid chromatography-mass spectrometry (LC-MS). Ion current measurements (i.e., MS1 ion currents) are used to quantify protein abundance. To improve proteome coverage, high-recovery sample preparation systems[9–11] and highly sensitive LC-MS systems[7,12,13] are usually employed. Although label-free approaches exhibit better quantification accuracy and higher dynamic range, their throughputs are limited, as each cell requires a >0.5 h-long LC-MS analysis. In the isobaric labeling approaches (e.g., tandem mass tags, TMT)[14–19], single-cell digests are labeled with unique isobaric tags that are then pooled together for a multiplex LC-MS analysis. Importantly, the peptides originating from different single cells appear as a single MS1 peak. As a consequence, the pooled ions contributing to a given precursor peak is higher than from individual cells and their fragmentations result in a richer MS2 spectrum for peptide identification. The released reporter ions infer protein abundance in different single cells. A "carrier" sample containing a larger amount of peptides than individual cells (e.g., ~100×) is spiked into each isobaric labeling pool to maximize the peptide identification (SCoPE-MS)[14,15,18]. Currently, the isobaric-labeling approaches have enabled the analysis of ~100 single cells per day. We anticipate that the throughput will increase gradually with new releases of higher multiplex isobaric reagents, shorter LC gradients, and the inclusion of ion mobility in single-cell proteomics pipelines.

Analogous to single-cell transcriptomics, microfluidic technologies play increasing roles in sample preparation for scProteomics[6,9,11]. By minimizing the sample processing volumes in nanowells or droplets, the non-specific-binding-related protein/peptide loss is reduced, resulting in improved sample recovery. More importantly, both protein and digestion enzyme concentrations (e.g., trypsin) increase in nanoliter volumes, enhancing digestion efficiency. For example, our lab developed a nanoPOTS (nanodroplet processing in one-pot for trace samples) platform for significantly improving proteomics sensitivity by minimizing the reaction volume to <200 nL[11]. NanoPOTS allowed reliably identifying 600–1000 proteins with label-free approaches[7,12,13]. When isobaric labeling approaches (SCoPE-MS) were used, ~1500 proteins could be quantified across 152 single cells and at a throughput of 77 cells per day[6,15]. Despite these innovations, challenges remain. In current microfluidic approaches, the sample processing volume is still >10,000 larger than a single cell. Gains would be expected from further miniaturizing the volumes, but it is presently constrained by liquid handling operations, including reagent dispensing, sample aspirating, transferring, and combination. Among these, the nanoliter-scale aspirating and transferring steps, which are commonly performed in isobaric-labeling workflows, are challenging, time-consuming, and prone to sample losses. Additionally, most reported microfluidic approaches employed home-built nanoliter liquid handling systems, which limits their broad dissemination.

Herein, we describe a nested nanoPOTS (N2) chip to improve the isobaric-labeling-based scProteomics workflow. Compared with our previous nanoPOTS chip[6,15], where nanowells are sparsely distributed, we cluster arrays of nanowells in dense areas and use them for digesting and labeling individual cells with single TMT sets. With the N2 chip, we eliminate the tedious and time-consuming TMT pooling steps. Instead, single-cell samples in one TMT set are pooled by simply adding a microliter droplet on top of the nested nanowell area and retrieving it for LC-MS analysis. The N2 chip reduces the sample processing volumes by one order of magnitude and allows over 5× more nanowells in one microchip for high-throughput single-cell preparation. We demonstrate the N2 chip not only efficiently streamlines the scProteomics workflow, but also improves sensitivity and reproducibility.

## Results

**Design and operation of the N2 chip.** The N2 chip is distinct from previous nanoPOTS chips[6,11,15,16]. We cluster an array of nanowells in high density and use each cluster for one multiplexed TMT experiment. In this proof-of-concept study, we designed 9 (3 × 3) nanowells in each cluster and 27 (3 × 9) clusters, resulting in a total of 243 nanowells on one chip (Figs. 1a and S1a). Additionally, we designed a hydrophilic ring surrounding the nested nanowells to confine the droplet position and facilitate the TMT pooling and retrieval steps. Compared with previous nanoPOTS chips[6,8,15], we reduced the nanowell diameters from 1.2 to 0.5 mm, corresponding to an 82% decrease in contact areas and an 85% decrease in total processing volumes (Table 1). The miniaturized volume resulted in a ~45× increase in trypsin digestion kinetics because both trypsin and protein concentrations were increased by 6.67×. Both the reduced contact area and increased digestion kinetics were expected to enhance scProteomics sensitivity and reproducibility.

The scProteomics sample preparation workflow using the N2 chip is illustrated in Fig. 1b. To sort single cells in the miniaturized nanowells, we employed an image-based single-cell isolation system (IBSCI, cellenONE F1.4). The cellenONE system also allowed us to dispense low nanoliter reagents for cell lysis, protein reduction, alkylation, and digestion. After protein digestion, TMT reagent is dispensed to label peptides in each nanowell uniquely. Finally, we distributed 10 ng boosting/carrier peptide and 0.5 ng reference peptide into each nanowell cluster to improve the protein identification rate (Fig. S1b)[14]. To integrate the N2 chip in our LC-MS workflow, we loaded the chip in a nanoPOTS autosampler[6]. We applied a 3 μL droplet on top of the nested nanowells, combined the TMT set, and extracted the peptide mixture for LC-MS analysis (Fig. 1b). Compared with our previous nanoPOTS-TMT workflow[6,15,16], the total processing time of each chip was reduced from 36.5 to 18 min (Fig. S1c), which is equivalent to the reduced time from 0.83 to 0.07 min for each single cell. As such, the N2 chip increases the single-cell processing throughput by >10×.

It should be noted that the N2 chip can be coupled with conventional LC systems without the use of the customized nanoPOTS autosampler. As shown in Fig. S1d, the user can manually add an 8-μL droplet inside the hydrophilic ring to pool the TMT-labeled single-cell samples and transfer it into an autosampler vial for LC injection. Recently, Schoof et al.[19] and Liang et al.[20] have demonstrated the Opentrons OT-2 liquid handler can reliably pipette low-μL-scale solutions for preparing single-cell samples. Similarly, the TMT pooling step for the N2 chip could be automated with conventional LC systems using the OT-2 robot.

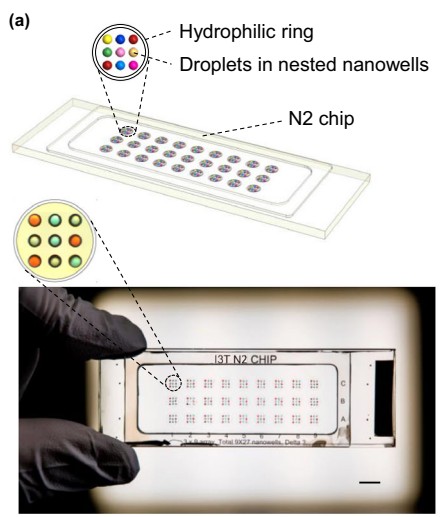 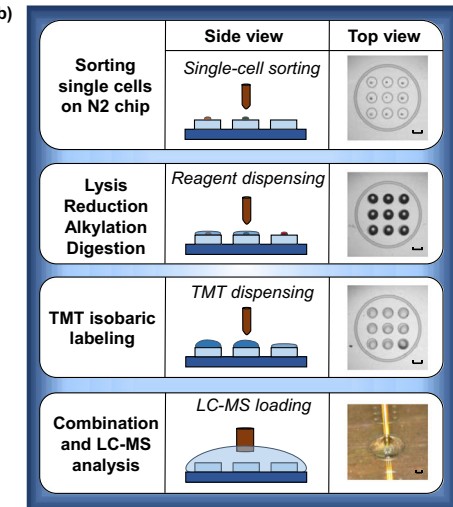

**Fig. 1 The design and operation of the nested nanoPOTS (N2) chip. a** A 3D illustration (top) and a photo (bottom) of the N2 chip. Nine nanowells are nested together and surrounded by a hydrophilic ring for a TMT set. The length of scale bar is 5 mm. **b** Single-cell proteomics workflow using the N2 chip. The length of scale bars is 0.5 mm.

**Table 1 Comparison of technical characteristics between N2 and nanowell chips.**

|  | N2 chip | Nanowell chip |
|---|---|---|
| Diameter (mm) | 0.5 | 1.2 |
| Contact area (mm$^2$) | 0.20 | 1.13 |
| Total volume (nL) | 30 | 200 |
| Digestion kinetics | 45× | 1× |
| Capacity (cells/chip) | 243 | 44 |
| Measured running time (min/chip, min/cell) | 18, 0.07 | 36.5, 0.83 |

**Sensitivity and reproducibility of the N2 chip.** We first benchmarked the performance of the N2 chip with our previous nanowell chip using diluted peptide samples from three murine cell lines (C10, Raw, SVEC). To mimic the scProteomics sample preparation process, we loaded 0.1 ng of peptide in each nanowell of both N2 and nanowell chips (Fig. S1b) and then incubated the chips at room temperature for 2 h. The long-time incubation would allow peptides to absorb on nanowell surfaces and lead to differential sample recoveries. The combined TMT samples were analyzed by the same LC-MS system. When containing at least one valid reporter ion value was considered as identified peptides, an average of 5706 peptides were identified with N2 chip, compared with only 4614 with nanowell chip. The increased peptide identifications result in a 15% improvement in proteome coverage; the average proteome identification number was increased from 1082 ± 22 using nanowell chips to 1246 ± 6 using N2 chips (Fig. 2a). We observed significant increases in protein intensities with N2 chips. The median log2-transformed protein intensities are 13.21 and 11.49 for N2 and nanowell chips, respectively, corresponding to ~230% improvement in protein recovery (Fig. 2b). Together, these results demonstrated that the N2 chips can improve sample recovery and proteomics sensitivity.

We assessed if the N2 chip could provide comparable or better quantitative performance compared with nanowell chips (Fig. S2a). As expected, more proteins are quantifiable with N2 chip when 70% valid values in each cell line were required; the quantifiable protein numbers were 870 and 1123 for nanowell and N2 chips, respectively (Fig. 2c). For nanowell chips, pairwise analysis of any two samples showed Pearson's correlation coefficients from 0.97 to 0.99 between the same cell types and from 0.87 to 0.95 between different cell types (Fig. S2b and S2c).

With N2 chips, Pearson's correlation coefficients were increased to a range of 0.98–0.99 for the same cell types, and a range of 0.91–0.96 for different cell types. We next evaluated the quantification reproducibility by measuring the coefficient of variations (CV) of samples from the same cell types. In intra-batch calculations, we obtained median protein CVs of <9.6% from N2 chips, which is more than two-fold lower than that from nanowell chips (median CVs of <24.9%) (Fig. 2d). Higher CVs were obtained between different TMT batches, which was known as TMT batch effect[21]. When Combat algorithm[22] was applied to remove the batch effect, the median protein CVs from N2 chip dropped to <6.7%. Such low CVs are comparable with bulk-scale TMT data, demonstrating the N2 chip could provide high reproducibility for robust protein quantification in single cells.

**Proteome coverage of single cells with the N2 chip.** We analyzed a total of 108 single cells (12 TMT sets) from three murine cell lines, including epithelial cells (C10), immune cells (Raw264.7), and endothelial cells (SVEC) (Fig. 3a). Noteworthily, these three cell types have different sizes, which allows us to evaluate if the workflow presents a bias in protein identification or quantification based on cell sizes. Specifically, Raw cells have a diameter of 8 μm, SVEC of 15 μm and C10 of 20 μm (Fig. S3a).

Among the 12 TMT sets, our platform identified an average of ~7369 unique peptides and ~1716 proteins from each set with at least one valid value in the nine single-cell channels (Fig. 3b, c). We identified a total of 2457 proteins, of which, 2407 proteins had reporter ion intensities in at least 1 single cells across the 108 cells (Fig. 3d). When a stringent criteria of >70% valid values was applied, the number of proteins dropped to 1437. As expected, we observed the numbers of proteins identified for three cell types ranked according to the cell sizes (Fig. S3a). An average of 1735, 1690, and 1725 proteins were identified in C10, RAW, and SVEC cells, respectively (Fig. 3e). Similar trends were also observed in the distribution of protein intensities (Fig. S3b).

Cheung and coworkers[17] recently introduced the software SCPCompanion to characterize the quality of the data generated from single-cell proteomics experiments employing isobaric stable isotope labels and a carrier proteome. SCPCompanion extracts signal-to-noise ratio (SNR) of single-cell channels and provides suggested cutoff values to filter out low-quality spectra to obtain high-quality protein quantitation. In line with our

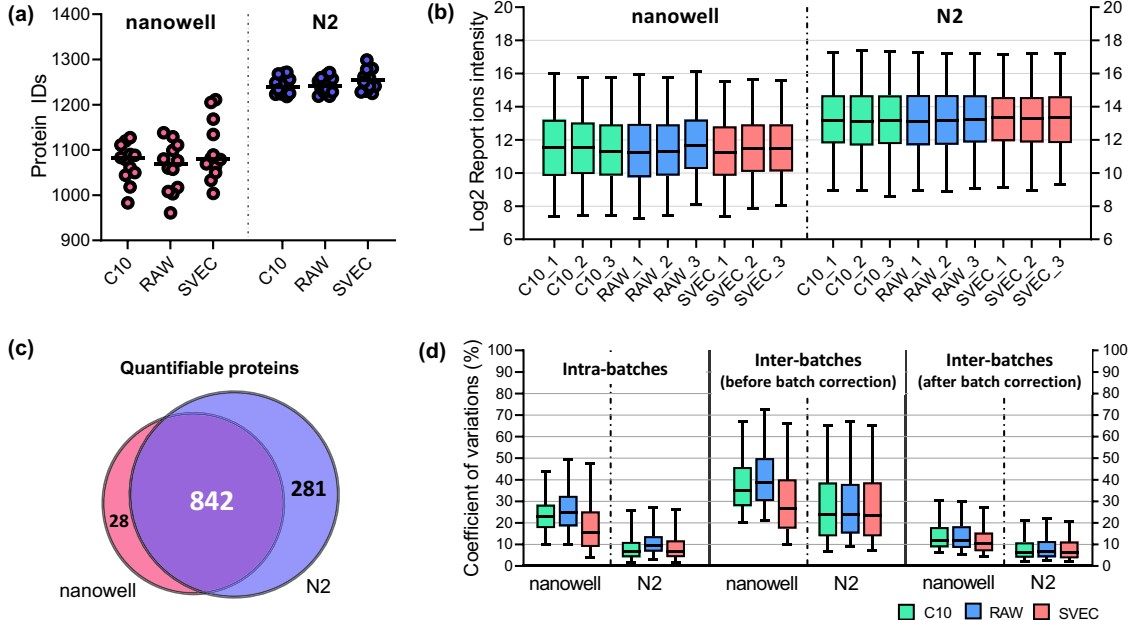

**Fig. 2 Performance comparison between nanowell chip and N2 chip. a** The numbers of protein identifications from 0.1 ng tryptic peptides from three cell lines (four TMT sets from nanowell chip and four TMT sets from N2 chip). **b** The distributions of log2 transformed protein intensities in each TMT channel ($n = 870$ proteins). **c** Venn diagram of quantifiable proteins between nanowell and N2 chips. (**d**) The distributions of the coefficient of variations (CVs) for proteins identified in each cell type. Protein CVs were calculated inside single TMT batches (left), among different TMT batches without batch corrections (middle), and with batch correction (right). From left to right, the number of proteins ($n$) are: 1005, 1002, 975, 1213, 1233, 1221, 745, 747, 759, 938, 927, 944, 736, 738, 747, 937, 924, and 940, respectively. In **b** and **d**, center lines show the medians; box limits indicate the 25th and 75th percentiles; whiskers extend 1.5 times the interquartile range from the 25th and 75th percentiles. Source data are provided.

experimental design, SCPCompanion estimated that ~0.1 ng proteins were contained in single cells, and the boost-to-single ratio is ~100 (Supplementary Data 1), indicating minimal peptide losses in the N2 chip. More importantly, the median SNR per single-cell sample was 14.4, which is very close to the suggested cutoff value of 15.5, corresponding to ~50% of raw MS/MS spectra can provide robust quantification. We also compared the data quality generated with previous nanowell chips and similar LC-MS setup[15,16]. The median SNR values per sample were 7.0[15] and 6.4[16], indicating the N2 chip-based workflow increased the SNRs by 106% and 125%, respectively (Fig. 3f). Recently, Hartlmayr et al.[23] observed that the use of TMTpro 16plex can give higher SNRs compared with TMT 10plex. To verify the performance improvement observed in the N2 chips was not solely due to the change of TMT reagents, we labeled single-cell-level peptides (0.1 ng) with both TMT 10plex and TMTpro 16plex. We analyzed them with the same MS using four different normalized HCD collision energy levels. As shown in Fig. S4b, MS1 peak intensities show similar distribution between the two TMT reagents. At MS2 level, we consistently observed TMTpro gave higher signal intensities (Fig. S4c) and SNRs (Fig. S4d), which agreed with the previous report[23]. The differences were much larger at lower HCD energy compared with high energy levels. The SNRs were increased by 212%, 119%, 67%, and 66% at HCD energies of 26%, 29%, 32%, and 35%, respectively. Because we used similar normalized HCD collision energy in our current N2 chip (34%) and previous nanowell chip-based work (35%), we reason the TMTpro reagent could lead to a similar improvement of ~66%, which accounts for ~40–50% of the total contributions.

**Cell typing with scProteomics**. To assess the quantitative performance of the N2 chip-based scProteomics platform, we first performed a pairwise correlation analysis using the 1437 proteins across the 108 single cells. As expected, higher correlations were

observed among the same types of cells and lower correlations among different types of cells (Fig. 4a). The median Pearson correlation coefficients are 0.98, 0.97, and 0.97 for C10, RAW, and SVEC cells, respectively. We next calculated the coefficient of variations (CVs) using protein abundances for the three cell populations. Interestingly, we see very low variations with median CVs <16.3% (Fig. S5), indicating protein expression is very stable for cultured cells under identical conditions. Principal component analysis (PCA) showed strong clustering of single cells based on cell types and the three clusters were well separated from one another (Fig. 4b). We compared these results to our previous PCA result obtained from the same three cell types using the nanowell-based platform (Fig. S6a)[16]. The median intra-cluster distances for the two-component PCA were relatively similar at 1.16 and 0.92 (median values) for nanowell and N2 chips, respectively (Fig. S6b). Conversely, the inter-cluster distances were 4.93 and 8.68 for nanowell and N2 chips, demonstrating the data generated from N2 chips have higher classification power for different cell populations.

To identify proteins leading the clustering of the three cell populations, an ANOVA test was performed (Permutation-based FDR < 0.05, S0 = 1). Of the total 1437 proteins, 1127 were significantly differentially changed in abundances across three cell types (Fig. 4c). Among them, 237 proteins were enriched in C10 cells, 203 proteins were enriched in SVEC cells, and 275 proteins were enriched in RAW cells. Proteins enriched in each cell type revealed differences in molecular pathways based on the REACTOME pathway analysis (Fig. S7). For example, the proteins higher in abundance in C10 cells were significantly enriched in REACTOME terms such as "vesicle-mediated transport", "membrane trafficking", "innate immune system", or "antigen processing-cross presentation". These functions are in line with the known functions of lung epithelial cells, of which the C10 are derived from[24]. The protein more abundant in RAW

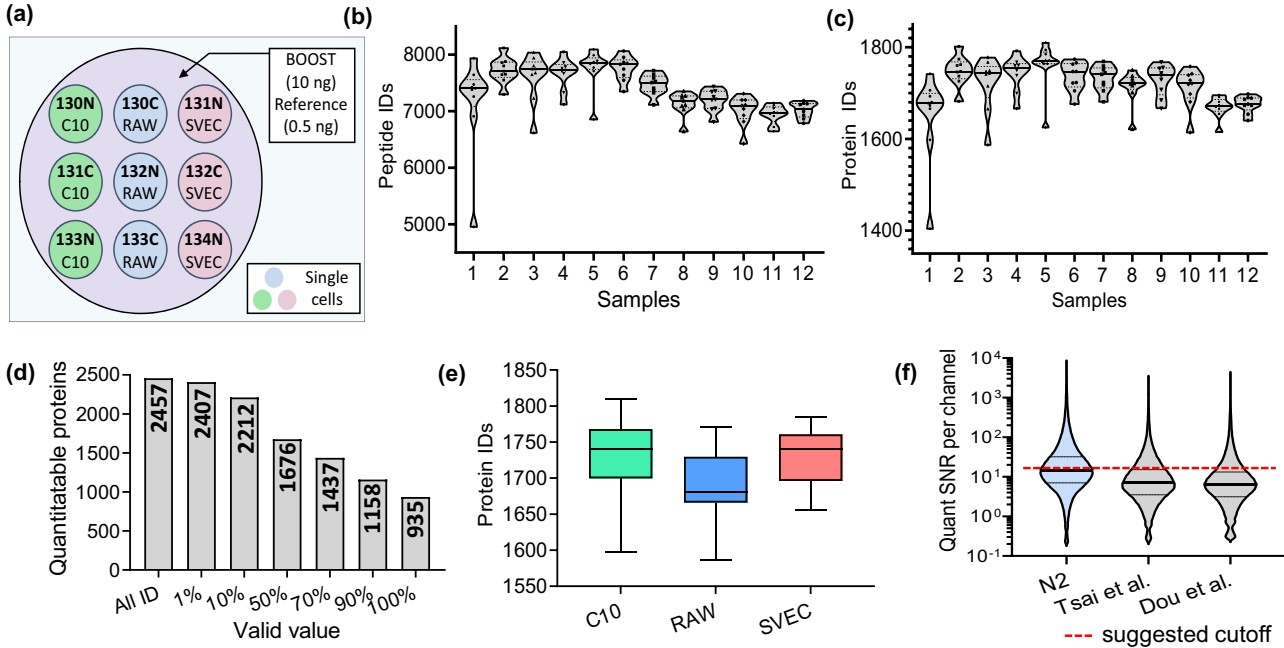

**Fig. 3 Single-cell proteomics with N2 chip. a** Experiment design showing single-cell isolation and TMT labeling on the N2 chip. **b, c** The numbers of identified peptides and proteins in 12 TMT sets. At least one valid value in the nine single-cell channels is required to count as an identification. Centerlines show the medians; top and bottom horizontal lines indicate the 25th and 75th percentiles, respectively. The data point (n) to generate the violin plots is 9. **d** The numbers of quantifiable proteins based on different percentages of required valid values. **e** Box plot showing the distributions of protein identification numbers (n = 36 single cells for each cell type). Centerlines show the medians; box limits indicate the 25th and 75th percentiles; whiskers extend 1.5 times the interquartile range from the 25th and 75th percentiles. **f** Violin plot showing the distributions of signal to noise ratio (SNR) per channel for raw single-cell signals calculated by SCPCompanion[17]. Two published TMT scProteomics datasets from our group using nanowell chip[15,16] were used to benchmark the data generated in this study. Centerlines show the medians; top and bottom horizontal lines indicate the 25th and 75th percentiles, respectively. The numbers (n) of SNR datapoints are 284,193 in N2 datasets, 282,190 in Tsai et al. datasets, and 260,039 in Dou et al. datasets. Source data are provided.

cells, which derive from murine bone marrow macrophages, were enriched in REACTOME terms associated with "neutrophil degranulation", "innate immune system" in line with their immune function. Other REACTOME terms related to the "ribosome" and the "pentose phosphate pathway" were also enriched. These pathways not only suggest that there is intricate cooperation between macrophages and neutrophils to orchestrate resolution of inflammation and immune system[25], but also show that system metabolism strongly interconnects with macrophage phenotype and function[26]. Proteins more abundant in SVEC cells (murine endothelial cells) were enriched in pathways, including "processing pre-mRNA", "cell cycle", or "G2/m checkpoints". This suggests its proliferation, migration, or coalescing of the endothelial cells to form primitive vascular labyrinths during angiogenesis[27].

**Identifying cell surface markers with scProteomics.** One of the unique advantages of scProteomics over single-cell transcriptomics is the capability to identify cell surface protein markers, which can be readily used to enrich selected cell populations for deep functional annotations. We assessed if we can use our scProteomics data to identify cell-type-specific membrane surface proteins for the three cell populations. We matched the enriched protein lists to a subcellular-localization database from UniProtKB, which consists of 2871 reviewed plasma membrane proteins for *Mus musculus* (updated on 01/04/2021). We generated a list containing 64 plasma membrane proteins (Supplementary Data 2). Among them, 17 proteins were highly expressed in C10 compared to RAW and SVEC cells, while 34 and 13 plasma membrane proteins were significantly enriched in RAW and SVEC cells, respectively. For example, NCAM1[28], EZRI[29],

and JAM1[30], which are previously known to protect the barrier function of respiratory epithelial cells by enhancing the cell-cell adhesion, are highly expressed in C10 cells (Fig. 5a, left panel). For RAW enriched membrane proteins, CD14[31] and CD68[31,32] are widely used as histochemical or cytochemical markers for inflammation-related macrophages (Fig. 5a, middle panel). CY24A is a sub-component of the superoxide generating NOX2 enzyme on macrophage membrane[33]. In terms of SEVC enriched protein markers, BST2 is known to be highly expressed in blood vessels throughout the body as an intrinsic immunity factor (Fig. 5a, right panel)[34]. Both of HMGB1 and DDX58 were found to be highly expressed in endothelial cells in lymph node tissue based on tissue microarray (TMA) results in human protein atlas. We also attempted to compare with our previous results using nanowell chips (Fig. S8)[16]. Only five out of the nine membrane proteins were significantly enriched in one of the cell types, and three were not detected, likely due to the lower sensitivity and reproducibility of the previous nanowell devices and workflows.

To evaluate the usability of scProteomics for identifying the cell-type-specific surface marker proteins, we selected one protein from each of the three cell populations (NCAM1_MOUSE fro C10; CD14_MOUSE for RAW; BST2_MOUSE for SVEC) and evaluated their specificity using an immunofluorescence imaging approach. As shown in Fig. 5b, immunofluorescence imaging validated the enrichments of the three marker proteins to their assigned cell types. It also confirmed their expected subcellular localization at the surface of the plasma membrane. Next, we assessed if these protein markers are specifically expressed in similar cell types in tissue samples. We verified the localization of the markers on human immunoperoxidase histology images generated by the Human Protein Atlas focusing on respiratory

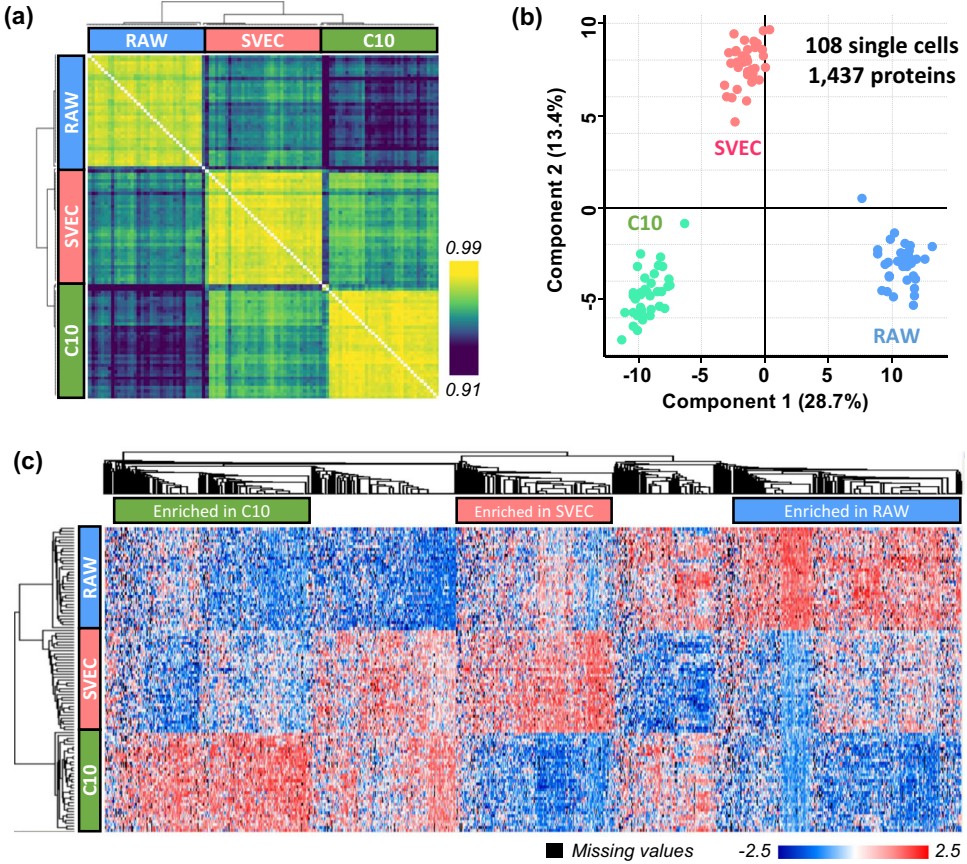

**Fig. 4 Evaluation of quantification performance for single cells. a** Clustering matrix showing Pearson correlations across 108 single cells using log2-transformed protein intensities. The color scale indicates the range of Pearson correlation coefficients. **b** PCA plot showing the clustering of single cells by cell types. Total 1437 proteins were used in the PCA projection. **c** Heatmap with hierarchical clustering showing 1127 significant proteins based on ANOVA test. Three protein clusters used for pathway analysis were labeled and highlighted. The color scale indicates the range of Z-score values. Source data are provided.

organs (lung, bronchi, and nasopharynx)[35]. While the general organization of the lung differs in human and mice (e.g.number of lobes, airway, and bronchi organization), the cell types composing the organ are almost identical as evidenced in a scRNA-seq study[36]. Thus, we speculated the human and mice cells should share many similarities in terms of protein expression patterns. As anticipated, the localization of the protein markers for similar cell types in human tissues is in agreement with our scProteomics data (e.g., C10 and RAW). Both EZRI and JAM1 enriched in C10 are localized in human epithelial cells (Fig. S9). The immune-cell-related markers, CD14, CD68, and CYBA (Uniprot protein name: CY24A_Human), are localized explicitly in macrophage cells in human lung tissues. Together, these results demonstrated cell-type-specific surface markers can be effectively identified by combining scProteomics with subcellular-localization information.

**Comparing scProteomics with scRNA-seq measurements**. We compared the scProteomics results to previously published scRNA-seq datasets containing 11 C10 cells[37] and 185 Raw cells[38] generated with SMART-Seq2 workflows. Compared with scRNA-seq, we observed higher Pearson correlation coefficients from scProteomics for both cell types (Fig. 6a). Specifically, the medians of correlation coefficients of mRNA abundances are 0.60 (C10) and 0.71 (RAW), while the coefficients of protein abundances significantly increased to 0.98 (C10) and 0.97 (RAW). The low variation in protein abundances can also be observed in the CV distributions of protein or mRNA abundances (Fig. 6b). Previous work have suggested moderate correlations between

protein and mRNA abundances of the same genes[39,40]. Our cross-correlation analysis between protein and mRNA shows similar trends with coefficients of 0.22 for C10 and 0.36 for RAW (Fig. 6c, d). These low correlations agree with bulk-scale measurement[41] and suggest scProteomics could provide additional information on the cell functions.

To identify the differentially expressed proteins and mRNAs between the two cell types, we performed t-test for both datasets. The proteins and mRNAs enriched either in C10 or RAW were moderately correlated. The overlaps between enriched mRNA and proteins were 44% for C10 and 40% for RAW cells (Fig. S10a). Most proteins and mRNAs followed a similar abundance pattern between the two cell types (Fig. 6e). The linear correlation coefficient of the log2(fold-changes) of the protein-mRNA pair is 0.55. The magnitude of the fold-change seemed higher for mRNA compared to protein (linear regression slope 0.08). This difference may indicate that a high amount of RNA is required to result in a moderate change of protein abundance. Another explanation is that the amplification steps employed in single-cell RNA sequencing may result in artificially inflated fold-changes[42]. Reactome pathway analysis for the significantly enriched proteins and mRNAs indicates general agreements between the two measurement types (Fig. S10b). However, a few enriched pathways were unique to either single-cell proteomics or RNA sequencing. For example, for pathways enriched in C10 cells, downstream signaling events of B cell receptor (BCR) was only detected at protein level and the adaptive immune system was only seen at mRNA level. For pathways enriched in RAW cells, the innate immune system was only observed at the protein level.

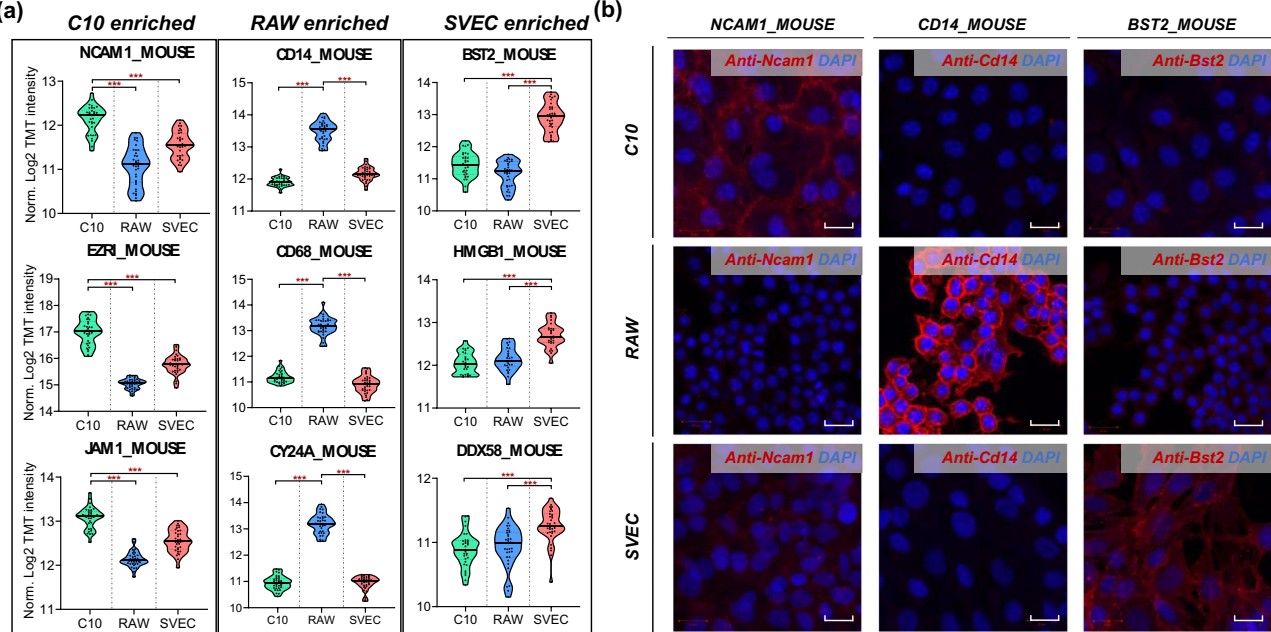

**Fig. 5 Prediction of cell-type-specific surface proteins with single-cell proteomics. a** Violin plots showing nine putative plasma membrane proteins enriched in three cell types. Proteins in each column are statistically significant (Two-sided t-test, ***p-value ***<0.001) expressed in the specific cell type. Centerlines show the medians; top and bottom horizontal lines indicate the 25th and 75th percentiles, respectively. For C10, $n = 34$ single cells; For RAW, $n = 35$ single cells; For SVEC, $n = 35$ SVEC single cells. **b** Immunofluorescence images showing the expression of NCAM1_MOUSE, CD14_MOUSE, and BST2_MOUSE in three cell populations. The protein abundance is visualized with red fluorescence, and DNA is visualized by DAPI staining (blue). The length of scale bar is 20 μm. Source data are provided.

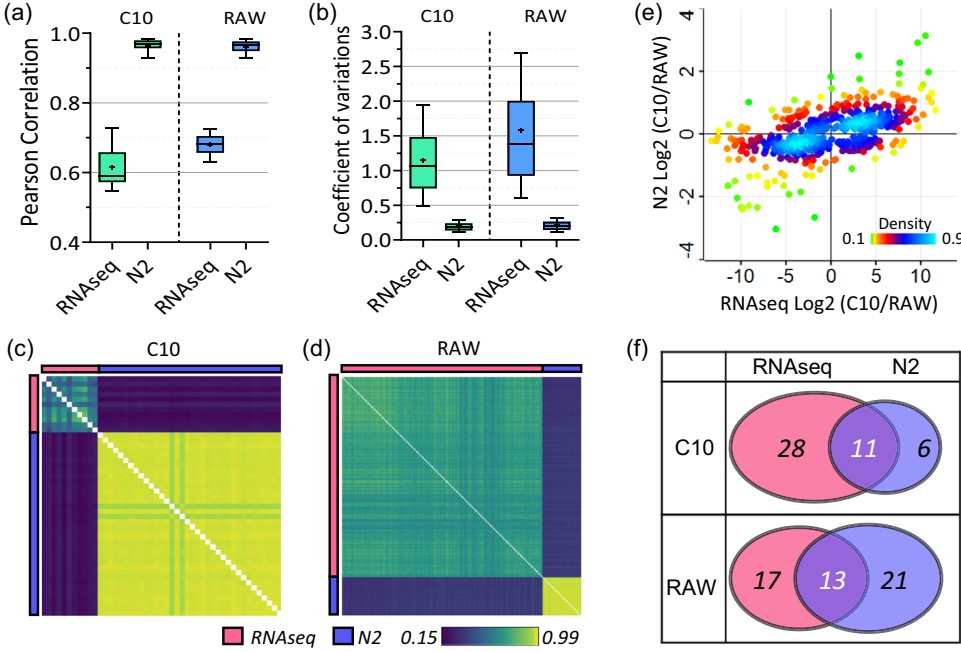

**Fig. 6 Integrative analysis of single-cell proteomics and single-cell transcriptomics datasets. a** Box plots showing the distributions of Pearson correlation coefficients and **b** coefficient of variations of transcript and protein abundances from scRNA-seq (Pearson correlation plot: $n = 11$ single C10 cells and 186 single RAW cells; coefficient of variation plot: $n = 1292$ genes for both C10 and RAW) and scProteomics (Pearson correlation plot: $n = 36$ single cells for both C10 and RAW; coefficient of variation plot: $n = 1292$ proteins for both C10 and RAW cells), respectively. Centerlines show the medians; box limits indicate the 25th and 75th percentiles; whiskers extend 1.5 times the interquartile range from the 25th and 75th percentiles. **c** Clustering matrix showing Pearson correlations of transcript and protein abundances for C10 and **d** RAW cells. The color scale indicates the range of Pearson correlation coefficients. **e** The linear correlation of log2-transformed fold changes of C10 and RAW cells between scRNA-seq and scProteomics. The color scale indicates the range of data point density. **f** Venn diagrams showing the overlap of membrane protein markers predicted by RNA seq and proteomics. Source data are provided.

However, several immune-related Reactome pathway terms were unique at mRNA level.

Finally, we assessed if mRNA and protein measurements predict the same membrane protein markers. After matching to the same subcellular-localization database, scRNA-seq measurements identified 30 membrane proteins for RAW cells and 40 proteins for C10 cells (Supplementary Data 2). The overlaps between the two measurements were moderate for both cell types (Fig. 6f). Less than 32.5% protein targets predicted by RNA-seq were found by proteomics measurements, indicating mRNA abundances cannot precisely infer membrane protein abundances. Interestingly, both protein and mRNA measurements identified the six proteins shown in Fig. 5 as significantly enriched markers. Overall, our analysis suggests the combination of the two modalities provides the most reliable membrane protein markers.

## Discussion

We have developed a high-throughput and streamlined scProteomics sample preparation workflow based on nested nanoPOTS (N2) chips. The N2 chips reduce nanowell volumes to ~30 nL and improve the protein/peptide sample recovery by 230% compared with our previous nanoPOTS chips[15,16]. The N2 design also significantly simplifies the TMT-based isobaric labeling workflow by eliminating the tedious sample pooling step (e.g., aspirating, transferring, and combining). With the N2 chip, 243 single cells can be analyzed in a single microchip, representing 5× more numbers than our previous chips.

In the near future, we envision the development of higher capacity N2 chips and/or stable isotope isobaric labeling reagents for hyperplexing scProteomics experiments (e.g., over 1000 cells per chip containing 5 × 5 array and 40 total clusters). A parallel droplet dispensing system could be developed for ultra-high throughput single-cell preparation. To further improve sample recovery and digestion efficiency, the nanowell diameters can be reduced to 0.2 mm or below, which corresponds to total droplet volumes of <5 nL. New microfluidic strategies could be developed to minimize droplet evaporation and increase droplet dispensing precision[43]. The proteome coverages could be improved with lower-flow LC systems[13,44], advanced MS instrumentation[23,40], and data analysis strategies[45].

Using a recently-developed software, SCPCompanion[17], we observed single-cell SNRs were greatly improved with N2 chip-based scProteomics workflow. Importantly, we observed high Pearson's correlation coefficient (~0.97) and low protein variations (median CVs of ~16.3%) in single cells, suggesting not only that the method is highly reproducible but also a low heterogeneity in cells cultured in favorable and identical conditions (e.g., nutrient-rich media without perturbation). These observations suggest cultured cells are good models to evaluate and benchmark the quantitative performance of scProteomics technologies.

Using three distinct cell lines, we verified that the scProteomics allowed to robustly classify cells based on their protein abundances and reveal functional differences between them. We also demonstrated the scProteomics allowed to directly identify cell surface markers by leveraging established subcellular-localization databases.

The direct comparison of our scProteomics with publicly available scRNA-seq datasets indicated that the amplitude of the variations observed in scProteomics was lower than those measured in transcriptomics. The reason for these discrepancies remains unclear as it may be real or the result of technical artifacts[42]. The integrative analysis showed moderate correlations between protein and transcript abundances, indicating

scProteomics can provide a complementary perspective on cellular states.

It should be noted that all the single-cell isolation and sample preparation were performed using a commercially available system (cellenONE). The microchip fabrication can be readily implemented in a typical cleanroom facility. Thus, we believe our N2 chip-based scProteomics workflow and related devices can be rapidly disseminated through a commercialization agreement.

We believe that the N2 chip scProteomics platform presented herein will enable the scientific community to study cell differentiation, tumor heterogeneity, and to identify rare cell populations from clinical specimens. Together with other technical developments, the N2 chip-based scProteomics platform could be extended to other functional proteomics measurements such as protein post-translational modifications, protein–protein interactions, and cell-specific proteoforms.

## Methods

**Fabrication and assembly of the N2 chips**. The chips were fabricated on glass slides using standard photolithography, wet etching, and silane treatment approach as described previously[11,46]. Briefly, as shown in Fig. 1 and S1a, 27 (3 × 9) nanowell clusters with a distance of 4.5 mm between adjacent clusters are designed on a single microscope slide (1 × 3 inches, Telic Company, Valencia, USA). In each cluster, nine nanowells with 0.5-mm diameter and 0.75-mm well-to-well distance are nested together. To facilitate droplet combination and retrieval process, a micro-ring surrounds the nested nanowells. After photoresist exposure, development, and chromium etching, the glass slide was etched to a depth of ~5 μm with buffered hydrofluoric acid[47]. The freshly etched slide was dried by heating it at 120 °C for 2 h and then treated with oxygen plasma for 3 min (AP-300, Nordson March, Concord, USA). To selectively pattern the chip, 2% (v/v) heptadecafluoro-1,1,2,2-tetrahydrodecyl-dimethylchlorosilane (PFDS, Gelest, Germany) in 2,2,4-trimethylpentane was applied on the chip surface and incubated for 30 min. After removing the remaining chromium layer, all the chromium-covered regions (nanowells and micro-rings) are hydrophilic, and exposed areas are hydrophobic. Finally, a glass frame was attached to the nanowell chip with epoxy to create a headspace for reaction incubation.

**Reagents and chemicals**. Urea, n-dodecyl-β-D-maltoside (DDM), Tris 2-carboxyethyl phosphine (TCEP), Iodoacetamide (IAA), Ammonium Bicarbonate (ABC), Triethylammonium bicarbonate (TEAB), Trifluoroacetic acid (TFA), Anhydrous acetonitrile (a-ACN), and Formic acid (FA) were obtained from Sigma (St. Louis, MO, USA). Trypsin (Promega, Madison, WI, USA) and Lys-C (Wako, Japan) were dissolved in 100 mM TEAB before usage. TMTpro 16plex, 50% hydroxylamine (HA), Calcein AM, Acetonitrile (ACN) with 0.1% of FA, and Water with 0.1% of FA (MS grade) were purchased from Thermo Fisher Scientific (Waltham, MA, USA).

**Cell culture**. Three murine cell lines (RAW 264.7, a macrophage cell line; C10, a respiratory epithelial cell line; SVEC, an endothelial cell line) were obtained from ATCC and cultured at 37 °C and 5% CO$_2$ in Dulbecco's Modified Eagle's Medium supplemented with 10% fetal bovine serum and 1× penicillin-streptomycin (Sigma, St. Louis, MO, USA). Three leukemia cell lines (MOLM-14, K562, and CMK) were kindly provided by Dr. Anupriya Agarwal at Oregon Health & Science University. MOLM-14 and K562 cells were grown in RPMI1640 medium supplemented with 10% FBS and 1× penicillin streptomycin, and CMK cells were maintained in RPMI-1640 medium supplemented with 20% FBS and 1× penicillin streptomycin.

**Bulk-scale proteomic sample preparation and mimic single-cell experiments**. The cultured cell lines were collected in a 15 mL tube and centrifuged at $1000 \times g$ for 3 min to remove the medium. Cell pellets were washed three times by 1× PBS buffer, then counted to obtain cell concentration. Ten million cells per cell population were lysed in a buffer containing 8 M urea in 50 mM ABC in ice. Protein concentration was measured with BCA assay. After protein was reduced and alkylated by DTT and IAA, Lys-C (enzyme-to-protein ratio of 1:40) was added and incubated for 4 h at 37 °C. Trypsin (enzyme-to-protein ratio of 1:20) was added and incubated overnight at 37 °C. The digested tryptic peptides were acidified with 0.1% TFA, desalted by C18 SPE column, and completely dried to remove the acidic buffer.

After measuring the peptide concentration with BCA assay, samples from three different cell types were mixed at 1:1:1 ratio and used for boost and reference samples. All peptide samples were dissolved with 50 mM HEPES (pH 8.5) followed by mixing with a TMT 10plex or TMT 16plex reagent in 100% ACN. To maintain high labeling efficiency, a TMT-to-peptide ratio of 4:1 (w/w) was used. After 1-h incubation at room temperature, the labeling reaction was terminated by adding 5% HA and incubating for 15 min. The TMT-labeled peptides were then acidified

with 0.1% FA and cleaned with C18 stage tips. Before use, different amounts of peptides (0.1 ng for mimic single cell, 0.5 ng for reference, 10 ng for boost) were diluted in 0.1% FA buffer containing 0.1% DDM (w/v) to prevent sample loss at low concentration conditions.

To mimic single-cell proteomics preparation in nanowell chips, 0.1 ng peptide samples in 200 nL buffer from the three cell lines were loaded into 1.2 mm nanowells using a nanoPOTS dispensing robot[11] and incubated for 2 h at room temperature. Next, samples from the same TMT set were collected and combined into a large-size microwell (2.2 mm diameter), which contained 10 and 0.5 ng TMT-labeled peptides for boost and reference samples, respectively.

To deposit these single-cell-level peptide samples to N2 chip, we employ a picoliter dispensing system (cellenONE F1.4, Cellenion, France) to dispense 0.1 ng peptide in 20 nL buffer in each nanowells (Fig. S1b). After incubating the chip at room temperature for 2 h, mixed boost and reference samples (10 ng and 0.5 ng, respectively) were equally distributed into each nanowell.

Samples in both nanowell chip and N2 chip were completely dried out in a vacuum desiccator and stored in a −20 °C freezer until analysis.

**ScProteomics sample preparation using the N2 chip**. The cellenONE system was used for both single-cell sorting and sample preparation on the N2 chip. Before cell sorting, all the cells were labeled with Calcein AM (Thermo Fisher) to gate out dead cells and cell debris. After single-cell deposition, 10 nL lysis buffer containing 0.1% DDM and 5 mM TCEP in 100 mM TEAB was dispensed into each nanowell. The N2 chip was incubated at 70 °C for 45 min in a humidity box to achieve complete cell lysis and protein reduction. Next, 5 nL of 20 mM IAA was added, followed by reaction incubation for 30 min in the dark. Proteins were digested to peptides by sequentially adding 0.25-ng Lys-C (5 nL) and 0.5-ng-trypsin (5 nL) into the nanowells and incubating for 3 h and 8 h, respectively. For isobaric labeling, we added 50 ng TMT tag in 10 nL ACN into each of the corresponding nanowells according to experimental design. An additional 10 nL, 100 mM TEAB buffer was added into each nanowell to compensate for the rapidly evaporated ACN solvent. After 1 h incubation at room temperature, the remaining TMT reagents were quenched by 5 nL of 5% HA. Finally, TMT labeled boost (10 ng) and reference (0.5 ng) peptides were distributed into nanowells. The samples were acidified with 5 nL of 5% FA and dried for long-term storage.

**LC-MS/MS analysis**. All the samples were analyzed with a nanoPOTS autosampler[6] equipped with a C18 SPE column (100 μm i.d., 4 cm, 300 Å C18 material, Phenomenex) and an LC column (50 μm i.d., 25 cm long, 1.7 μm, 130 Å, Waters) heated at 50 °C using AgileSleeve column heater (Analytical Sales and Services Inc., Flanders, NJ). Dried samples from nanowell chips or N2 chips were dissolved with Buffer A (0.1% FA in water), then trapped on the SPE column for 5 min. Samples were eluted out from the column using a 120-min gradient from 8% to 45% Buffer B (0.1% FA in ACN) and a 100 nL/min flow rate.

An Orbitrap Eclipse Tribrid MS (Thermo Scientific, Xcalibur Ver. 4.3.73.11) operated in data-dependent acquisition mode was employed for all analyses for peptides. Peptides were ionized by applying a voltage of 2200 V and collected into an ion transfer tube at 200 °C. Precursor ions from 400 to 1800 $m/z$ were scanned at 120,000 resolution with an ion injection time (IT) of 118 ms and an automatic gain control (AGC) target of 1E6. During a cycle time of 3 s, precursor ions with > +2 charges and >2E4 intensities were isolated with a window of 0.7 $m/z$, an AGC target of 1E6, and an IT of 246 ms. The isolated ions were fragmented by a higher energy collisional dissociation (HCD) level of 34%, and the fragments were scanned in an Orbitrap at 120,000 resolution.

A Q-Exactive plus MS (Thermo Scientific, Xcalibur Ver. 4.0.27.19) was used to analyze TMT 10pelx and TMT 16plex-labeled peptide samples. The MS1 spectra were collected in Orbitrap at a scan range of 400–1800 $m/z$, a resolution of 35,000 and an AGC target of 3E6. Top-10 precursor ions with intensities of >3E5 and chargers of >2+ were selected for fragmentation with HCD levels from 26 to 35%, an AGC target of 5E6, and a maximum IT of 300 ms. The fragments were scanned in an Orbitrap at 70,000 resolution.

**Database searching**. All the raw files from the Thermo MS were processed by MaxQuant[48] (Ver. 1.6.14.0) with the *UniProtKB* protein sequence database of *Mus musculus* species (downloaded on 05/19/2020 containing 17,037 reviewed protein sequences). Reporter ion MS2 was set as the search type and TMT channel correction factors from the vendor were applied. The mass tolerances for precursor ions and fragment ions were set as 4.5 ppm and 20 ppm, respectively in MaxQuant. Specific digestion enzymes were set as Trypsin and LysC. The number of allowed missed cleavages was set as 2. The match tolerance, de novo tolerance, and deisotoping tolerance for MS/MS search were 20, 10, and 7 ppm, respectively. The minimum peptide length was six amino acids and the maximum peptide mass was 4600 Da. Protein acetylation in N-terminal and oxidation at methionine were chosen as variable modifications, and protein carbamidomethylation in cysteine residues was set as fixed modification. Both peptides and proteins were filtered with false discovery rates (FDR) of <1% to ensure identification confidence.

**Single-cell proteomics data analysis**. SCPCompanion (Ver 15.0, https://www.github.com/scp-ms/SCPCompanion) was used to access the data quality by

extracting the summed signal-to-noise ratio (SNR) of single-cell channels[17]. The corrected reporter ion intensities from MaxQuant were imported into Perseus (Ver. 1.6.14.0)[49] and were log2-transformed after filtering out the reverse and contaminant proteins. Proteins containing >70% valid values in each cell type were considered quantifiable. Missing values were imputed based on a standard distribution of the valid values (width: 0.3, downshift: 1.8). The summed reporter ion intensities of the quantifiable proteins were normalized using quantile normalization method. To correct the batch effect from multiple TMT sets, we used the SVA Combat algorithm[22], which is embedded in Perseus. Next, the data matrix was separated by cell types and grouped by TMT channel. Combat algorithm was also applied to minimize the TMT channel effect. The combined matrix was then used for statistical analysis, including principal component analysis (PCA) and hierarchical clustering analysis. ANOVA tests were performed to determine the proteins showing statistically significant abundance differences across the three cell types (Permutation-based FDR < 0.05, $S_0 = 1$), and a two-way student $t$-test was applied to explain the significant differences between two groups ($p$-value < 0.05). The processed data were visualized with Graphpad (Prism Ver 8.3.0) and Perseus.

Protein intensities without missing values in each cell type in intra-batch or inter-batches were used to calculate the coefficient of variations (CVs) in Excel (Microsoft office 365). Briefly, for intra-batches, the CVs were calculated using raw protein intensities inside each TMT set and then pooled together to generate the box plots. For inter-batches without batch correction, the CVs were calculated using raw protein intensities across all the TMT sets. To calculate the CVs of intra-batches with batch corrections, raw protein intensities were log2 transformed and missing values were imputed. After normalization and batch correction using Combat algorithm[22], proteins with imputed values were replaced to 'NaN' and filtered out. The protein intensities were exponentially transformed to calculate the CVs.

The Reactome pathway analysis was conducted on the STRING-db tool (Ver. 11.0b, https://version-11-0b.string-db.org/). Briefly, cell-type-specific regulated proteins with statistical significance were submitted in the multiple proteins windows and selected an organism of *Mus Musculus*. The results of Reactome pathways were exported with matched protein and gene lists and FDR values.

**Immunofluorescence staining**. Washed cells in PBS were fixed in fresh 4% paraformaldehyde (PFA) for 10 min at room temperature and quenched with 0.1% sodium borohydride for 7 min to get rid of free aldehyde groups for preventing autofluorescence of cells. The cells were then permeabilized and blocked using 1% BSA in PBST (PBS with 0.05% Tween 20) to minimize nonspecific binding of the antibodies. Three recombinant antibodies (Anti-NCAM1, ab220360; Anti-CD14, ab221678; Anti-BST2, ab246508) were purchased from Abcam (Cambridge, MA, USA). The cells were incubated in the diluted primary antibodies (1:1000 anti-CD14, 1:2000 anti-BST2 and 1:1000 anti-NCAM1) overnight at 4 °C in 1× PBS with 1% BSA. Subsequently, the labeled cells were washed with ice-cold PBS followed by incubating with Alexa Flour 546 goat anti-rabbit IgG (Invitrogen Cat#A11010) at 1:1000 dilution in 1× PBS with 1% BSA for an hour at room temperature in the dark. Immunostaining images were visualized with an inverted confocal fluorescence microscope (Zeiss LSM 710) with a 63× objective (NA 0.75). The DAPI nuclear stain was excited by a 405 nm wavelength laser. The dye conjugated antibodies (Alexa Flour 546) were excited by 561 nm wavelength laser. The antibody fluorescence channel, DAPI fluorescence channel, and bright-field channel were acquired simultaneously. The Zeiss imaging software ZEN (2.1 SP2 version 130.2.518) was used to control the microscope, acquire the data, and export the images.

**Integrative proteomics and transcriptomics analysis**. Single-cell transcriptomics datasets containing transcript abundance of 11 C10[37] and 186 RAW[38] cells were re-analyzed. Both datasets were generated with Smart-Seq2 protocols. The data of C10 cells was obtained with unit of counts, while data of RAW264.7 cells was obtained with unit of fragments per million (FPM). To allow comparisons, we converted the data of C10 cells with unit of FPM using a python package bioinfokit (https://github.com/reneshbedre/bioinfokit)[50]. Next, we transformed scRNASeq data into log2 scale. After normalization and scaling, we selected the genes commonly captured across all transcriptomic and proteomic datasets for further correlation and other comparative analyses.

**Reporting summary**. Further information on research design is available in the Nature Research Reporting Summary linked to this article.

## Data availability
The mass spectrometry raw data have been deposited to the ProteomeXchange Consortium via the MassIVE partner repository with the dataset identifier MSV000086809 and are available at https://doi.org/10.25345/C5JR4K or ftp://massive.ucsd.edu/MSV000086809/. Single-cell RNA sequencing data for control RAW cells[38] were downloaded from NCBI Gene Expression Omnibus with accession number GSE94383. Single-cell RNA sequencing data for C10 cells[37] were requested from the author and are provided in Supplementary Data 3. Source data are provided with this paper.

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

## Acknowledgements

We thank Matthew Monroe for helping with data deposition. We thank PNNL photographer Andrea Starr for taking the chip photo (Fig. 1a). We thank Dr. Anupriya Agarwal at Oregon Health & Science University for providing us the leukemia cell lines. This work was supported by a Laboratory Directed Research and Development award (Y.Z.) from Pacific Northwest National Laboratory and an Intramural program (Y.Z.) at EMSL (grid.436923.9), a DOE Office of Science User Facility sponsored by the Office of Biological and Environmental Research and operated under Contract No. DE-AC05-76RL01830. Part of this work is also supported by NIH grants U01 HL122703, U01 HL148860 (J.N.A. and G.C.C.) and P41 GM103493 (R.D.S.).

## Author contributions

Y.Z conceptualized and designed the research. S.M.W., V.A.V. and H.S.M. fabricated N2 microchips. J.W., R.L.S., L.M.M., C.F.T. and J.C.B. performed the cell culture and sample

preparation. S.M.W. and R.J.M performed LC-MS analysis. J.W., L.M.M., S.F., C.F.T., T.L., G.C. and Y.Z. analyzed data. J.W., C.F.T., J.N.A., G.C., R.D.S., L.P.T. and Y.Z. wrote the manuscript.

## Competing interests

J.C.B. is an employee of Scienion. Battelle Memorial Institute has submitted a U.S. provisional patent application (Application number: 63/150,824; Inventors: Ying Zhu, Jongmin Woo, and Ljiljana Pasa-Tolic; Status of application: Pending). The patent covered the design of nested nanoPOTS devices and the associated operation methods. Other authors declare no competing interests.
