## [Peer Review File · Nature Communications]

REVIEWER COMMENTS

Reviewer #1 (Remarks to the Author):

Woo et al describe further developments of their previously introduced microfluidic single cell proteomics platform. The technology is without doubt relevant for the community, given the increasing interest in single-cell multi-omics. The authors very systematically describe the (quantitative) improvements in technical performance, always in direct comparison to their previous work. However, a truly disruptive step forward or entirely new concepts are missing. Furthermore the authors don't provide a benefit in terms of biological insights, e.g. an application that would not have been possible with any of the previous systems. Instead, we only learn that all specific (technical) parameters of the system have been improved by x%. One could simply ask why a further reduction of the nanowell size would not lead to even better performance. Similar thoughts are discussed in the discussion part, making me believe that what is described here might just be an intermediate step. What are the ultimate limits? What are disruptive concepts that may allow to overcome these?

Further points:

- In line with demonstrating biologically meaningful improvements, the authors should discuss more functional readouts. For example, they could at least speculate on applications in phosphoproteomics (e.g. in the discussion part) and the detection of other post translational modifications.
- Similarly, a direct comparison of the current data with scRNAseq data of the exact same cell types (e.g. from public data bases) would be interesting. What relevant additional information could one obtain with the described N2 platform?
- Old references 6 and 15, 16 seem to describe their core platform already previously. The additional implementation of a nested approach only increased the throughput from 77-152 singles cells to 243 cells, while similar protein numbers were detected (about 1500 after quality filtering). This corresponds to a rather incremental improvement. No new working principles or disrupting concept are introduced, but mainly just smaller nanowell diameters and slightly higher overall numbers of wells.
- The PCA plots in Figures 4b (new platform) and S5A (old platform) are not elementary different in terms of content. In both cases the three different cell types form separate distinguishable clusters. What new applications are enabled by the technical improvements – could one for example distinguish activated T-cell from non-activated T-cells?

In summary, all experiments were performed very systematically and the paper is nicely written, but it remains a bit elusive how meaningful the technical improvements are in terms of new biological applications that could be addressed.

Reviewer #2 (Remarks to the Author):

Summary: The authors present an updated version of their nanoPOTS approach by introducing the N2 chip design and coupling it to a commercial cell sorter/liquid handler (cellenOne). The updated sample preparation workflow further decreases surface contacts, increases the number of cells that can be prepared on one chip, and ultimately increases the signal detected within the mass spectrometer. The authors demonstrate the improved proteomic depth and show an example of how single cell proteomics can be used to identify cell specific surface markers. Overall, the paper is technically sound and demonstrates an important step forward for the field with the inclusion of a commercially available liquid handling system into their workflow. Previously, this had been a bottleneck to the expansion of the nanoPOTS technique. Due to these advances, this manuscript represents an important report to the field of single cell proteomics and demonstrates a path toward tackling one of the largest challenges – sample throughput. There are only minor comments surrounding the claims of increased signal to noise ratio (SNR) and the biological significance of the identified cell surface markers. However, these can likely be addressed in review, after which the manuscript would be acceptable for publication.

Minor Points:

1. Line 243: The authors state that the N2 chip can be directly coupled with conventional LC system, but recommend adding 8- μ L and aspirating into an autosampler vial. Is this meant to be manual? Or are the authors suggesting there is an automated way to do this?

2. Figure 3f: For the SNR increases for the N2 chip vs. Tsai et al and Dou et al – what is the contribution of the new TMT tag structure to this increase in signal? The TMTpro reagents fragment more easily and produce higher reporter ion signal. It's possible that the increase in SNR is due primarily to the new tag structure and not the advances of the N2 chip. Perhaps the authors could demonstrate that their collision settings for TMTpro generate similar levels of TMT reporter ion signal as compared to the previous settings used for TMT tags.

3. Cell surface markers with scProteomics: The authors demonstrate that they were able to detect membrane proteins that were specific for one cell type as compared to the other two cell types. However, I wonder how specific these protein markers would be compared to various tissues within a mouse.

There has been previously catalogued proteomes of various mouse tissues. It would be great to understand if these membrane proteins has been detected specifically in one of those mouse tissues because it would hint at a usable specificity beyond this three cell system.

Likewise, if there is RNAseq data available for these cells – would these markers have been predicted by the RNAseq data? The real value for scProteomics would be if these markers were not revealed by the RNAseq analysis.

Reviewer #3 (Remarks to the Author):

Summary:

The authors report the nested nanoPOTS (N2) chip for higher throughput and sensitivity over the group's previously reported nanoPOTS platform. The N2 chip demonstrates formidable improvements in sample preparation for single-cell mass spectrometry (e.g., >10x improvement in throughput, 15% improvement in proteome coverage, and ~230% improvement in protein recovery compared to the previous nanoPOTS design) by employing a single cell and nanoliter volume dispensing instrument (cellenONE). This work has the potential to substantially influence the field of single-cell analysis as researchers turn their head toward single-cell proteomics as a rising competitor and complementary technology to single-cell genomics. Moreover, by using commercially and publicly available equipment and software, the authors prime the nanoPOTS technology for broader adoption. Overall, this manuscript provides an excellent introduction to the field of single-cell mass spectrometry for a general scientific audience, is well written and organized, provides sufficient detail for reproducibility, and appropriate statistical analysis. Therefore, this publication is well suited for publication in Nature Communications after the following minor comments are addressed.

Minor Comments:

1. Abstract, page 2, line 22: Define TMT when it first appears.
2. Figure 1a description, page 5, line 82: “an TMT set” should be “a TMT set”
3. Methods, page 6, line 97: “incubate” should be “incubated”
4. Results, page 19, lines 382-383: Make it more clear here that the low sensitivity and reproducibility is referring to data acquired with the previously published nanowell chip and that the N2 chip offers an advantage. Currently, it reads as if the N2 chip has low sensitivity and reproducibility.

Point-by-Point Response to the Reviewer Comments

Reviewer comments in italics, responses in blue.

Reviewer: #1

Woo et al. describe further developments of their previously introduced microfluidic single-cell proteomics platform. The technology is without doubt relevant for the community, given the increasing interest in single-cell multi-omics. The authors very systematically describe the (quantitative) improvements in technical performance, always in direct comparison to their previous work. However, a truly disruptive step forward or entirely new concepts are missing. Furthermore the authors don't provide a benefit in terms of biological insights, e.g. an application that would not have been possible with any of the previous systems. Instead, we only learn that all specific (technical) parameters of the system have been improved by x%.

We thank the Reviewer for the time committed to help us improve the manuscript. We are pleased the Reviewer recognizes the relevance of our most recent technology advancements for broader implementation of scProteomics.

As pointed out by several recent publications (e.g., Cheung et al., *Nat. Methods*, 2021; Ctortocka et al., *Anal. Sci. Adv.*, 2021; Kelly et al., *Mol. Cell. Proteom.*, 2021), single-cell proteomics is still in its early stages of development and many challenges remain, including poor sample recovery, low throughput, reproducibility and robustness. Our nested nanoPOTS (N2) platform addresses several of these challenges and significantly improves the overall analytical performance of scProteomics. We believe the N2 design represents a critical breakthrough that will move the field forward and enable many previously intractable applications, as has been clearly recognized by the Reviewers 2 and 3. The N2 design not only improves sample recovery by reducing the processing volume, but also significantly increases the system robustness and measurement throughput by eliminating the tedious and time-consuming pooling of TMT channels. Perhaps even more importantly, the implementation of our scProteomics workflow on the commercially available system (CellenONE) signifies a transformational step for scProteomics as it allows broad dissemination of this enabling technology.

We agree the inclusion of a specific biological application as a demonstration of the transformative nature of our technology would have strengthened the manuscript. While we would have liked to comply with the Reviewer's request, the Covid-19-related shutdown of research labs slowed down our research activities and prevented us to properly address this issue. Rapid dissemination of our technology to the broader scientific community has always been and remains our goal. We believe the N2 design coupled with CellenONE platform as described in the revised manuscript accomplishes this goal and as such represents "a truly disruptive step forward". We have initiated several collaborative efforts whereby we are applying our novel scProteomic workflow to address the most challenging biological questions regarding cell typing, toxin response heterogeneity, and cell differentiation. As indeed recognized by the Reviewer, these studies will take months to be completed. This manuscript accomplishes an

important goal to quickly disseminate new advancements and inspire other scientists to further develop and broadly apply scProteomics. We will include many more applications of the N2 technology in our future studies.

One could simply ask why a further reduction of the nanowell size would not lead to even better performance. Similar thoughts are discussed in the discussion part, making me believe that what is described here might just be an intermediate step. What are the ultimate limits? What are disruptive concepts that may allow to overcome these?

We agree with the Reviewer's comment here. Further reduction in nanowell size would lead to improved sample recovery and overall better performance. This is, however, a technically challenging endeavor, requiring novel approaches to minimize the evaporation of smaller droplets and further improvements of the droplet dispensing precision. Current implementation with 0.5 mm nanowell and ~30 nL total reaction volume balances out system robustness and attainable analytical performance. With current technology, it might be possible to reduce the nanowell size to ~0.25 mm and a total reaction volume to <5 nL, yielding an incremental improvement but not a major leap forward. We thank the Reviewer for bringing this to our attention and have included a brief paragraph clarifying these concerns in the conclusion section of the revised manuscript.

Further points:

- *In line with demonstrating biologically meaningful improvements, the authors should discuss more functional readouts. For example, they could at least speculate on applications in phosphoproteomics (e.g. in the discussion part) and the detection of other post translational modifications.*

Functional proteomics measurements such as protein post-translational modifications, proteoforms, and protein-protein interactions are arguably the most exciting advantages of scProteomics relative to single-cell sequencing technologies. We have expanded the conclusion section of the manuscript to emphasize these arguments to read as

“Together with other technical developments, the N2 chip-based scProteomics platform could be extended to other functional proteomics measurements such as protein post-translational modifications, protein-protein interactions, and cell-specific proteoforms.”

Similarly, a direct comparison of the current data with scRNAseq data of the exact same cell types (e.g. from public data bases) would be interesting. What relevant additional information could one obtain with the described N2 platform?

We thank the Reviewer for another great suggestion. To address this question, we re-analyzed two publicly available scRNA-seq datasets for C10 and RAW cells generated by the SMART-seq2 protocol (Figure 6). Our analysis suggests protein abundances are more stable across single

cells compared with mRNA abundances. We observed that the median Pearson correlation coefficients of mRNA abundances were below 0.71, while the median coefficients for proteins were above 0.97%. Similarly, the median CVs of mRNA abundance were higher than 100%, while protein abundance CVs were below 16.3%. We also observed a poor cross-correlation between protein and mRNA abundances with Pearson correlation coefficients of <0.36 . These results suggest scProteomics provides additional and complementary information for understanding the molecular underpinning of the cellular functions.

We also performed Reactome pathway analysis of enriched proteins and mRNAs between the two cell types. Although enriched pathways generally agreed between the two measurement types, we also observed pathways enriched solely either in proteomics or in transcriptomics datasets. For example, for pathways enriched in C10 cells, downstream signaling events of B cell receptor (BCR) was only detected at proteome level and the adaptive immune system was only detected at transcriptome level. For pathways enriched in RAW cells, the innate immune system was only observed at the proteome level. However, more immune-related pathways were detected at the transcriptome level, suggesting that some mRNA modulations are invisible at the protein level and vice versa.

In the revised manuscript, we added a new Figure 6 and accompanying text describing the comparison of scRNA-seq and scProteomics datasets.

Figure 6. (a) Box plots showing the distributions of Pearson correlation coefficients and (b) coefficient of variations of transcript and protein abundances from scRNA-seq and scProteomics, respectively. Centerlines show the medians; box limits indicate the 25th and 75th percentiles; whiskers extend 1.5 times the interquartile range from the 25th and 75th percentiles. (c) Clustering matrix showing Pearson correlations of transcript and protein abundances for C10 and (d) RAW cells. (e) The linear correlation of log2-transformed fold changes of C10 and

RAW cells between scRNA-seq and scProteomics. (f) Venn diagrams showing the overlap of membrane protein markers predicted by RNA seq and proteomics.

Old references 6 and 15, 16 seem to describe their core platform already previously. The additional implementation of a nested approach only increased the throughput from 77-152 singles cells to 243 cells, while similar protein numbers were detected (about 1500 after quality filtering). This corresponds to a rather incremental improvement. No new working principles or disrupting concept are introduced, but mainly just smaller nanowell diameters and slightly higher overall numbers of wells.

To further clarify the difference between the original nanowell chip and the new N2 chip, we included a photograph of the original nanowell chip as Figure S2a. The original chip can process only 44 single cells (corresponding to four TMT sets). Hence, four chips would be required to analyze ~150 cells, and nanoliter-scale volumes make the pooling of the TMT-labeled single-cell samples technically challenging. The N2 design not only allows for the processing of ~150 cells simultaneously but also streamlines the sample handling by simply using a large droplet to pool the samples.

As pointed out by the Reviewer, we did not observe a significant increase in proteome coverage. This is likely because the coverage remains to be determined by the 100× carrier/boosting channel, which provides rich fragmentation information for confident peptide identification. Major improvements brought by the N2 chip are the quantification precision as indicated in Figures 2 and 3, and improved sample processing throughput and robustness as indicated in Table 1.

Original nanoPOTS chip

Nested nanoPOTS chip

Figure S2. The original nanoPOTS chip used in ref 6 and 15, 16 (top) and the new nested nanoPOTS chip used in the current work (bottom).

The PCA plots in Figures 4b (new platform) and S5A (old platform) are not elementary different in terms of content. In both cases the three different cell types form separate distinguishable clusters. What new applications are enabled by the technical improvements – could one for example distinguish activated T-cell from non-activated T-cells?

Although both PCA plots indicated clustering by cell type, the classification power of the N2 platform is significantly higher as shown in Figure S6b. The inter-cluster distances increased from 4.93 to 8.68. We believe the improved quantification will allow us to describe subtle differences in protein abundances to identify rare cell types, which may lead to new biological insights.

In summary, all experiments were performed very systematically and the paper is nicely written, but it remains a bit elusive how meaningful the technical improvements are in terms of new biological applications that could be addressed.

As highlighted above, the main objective of this manuscript is to share technological advances leading to improved sensitivity, throughput and robustness of the microfluidics-based scProteomics platform. We also aim to broadly disseminate new technology through the implementation on a commercially available platform. Since the accessibility to innovative technologies has been the main limitation for widespread usage of microfluidics methods, we believe our efforts represent “a truly disruptive step forward” that will be of interest to the *Nature Communications* audience.

Reviewer #2:

The authors present an updated version of their nanoPOTS approach by introducing the N2 chip design and coupling it to a commercial cell sorter/liquid handler (cellenOne). The updated sample preparation workflow further decreases surface contacts, increases the number of cells that can be prepared on one chip, and ultimately increases the signal detected within the mass spectrometer. The authors demonstrate the improved proteomic depth and show an example of how single cell proteomics can be used to identify cell specific surface markers. Overall, the paper is technically sound and demonstrates an important step forward for the field with the inclusion of a commercially available liquid handling system into their workflow. Previously, this had been a bottleneck to the expansion of the nanoPOTS technique. Due to these advances, this manuscript represents an important report to the field of single cell proteomics and demonstrates a path toward tackling one of the largest challenges – sample throughput. There are only minor comments (1) surrounding the claims of increased signal to noise ratio (SNR) and (2) the biological significance of the identified cell surface markers. However, these can likely be addressed in review, after which the manuscript would be acceptable for publication.

We appreciate the Reviewer’s recognition of the novelty of our work and have endeavored to address all the Reviewer’s comments below.

Minor Points: 1. Line 243: The authors state that the N2 chip can be directly coupled with conventional LC system, but recommend adding 8- μ L and aspirating into an autosampler vial. Is this meant to be manual? Or are the authors suggesting there is an automated way to do this?

We thank the Reviewer for raising this important point. We recommended to pool the TMT-labeled single-cell samples with a micropipette as the 8- μ L droplet can be manipulated manually (Figure S1d). The hydrophilic ring surrounding the nanowell array can effectively confine the

droplet. To automate the process and increase throughput, we suggest using an Opentrons OT-2 liquid handler. Recent studies by Schoof et al. (*Nat. commun.*, 2021) and Liang et al. (*Anal. Chem.*, 2021) have demonstrated OT-2 can reliably pipette low- μ L solution for single-cell preparation. OT-2 robot allows the use of customized microwell plates (or microchips) and precise calibration of the microwell position before pipetting. A commercially available slide holder (*e.g.*, Thorlabs) can be used to mount the N2 chip for reproducible alignment. We have clarified and expanded the relevant section in the revised manuscript to read as follows:

“As shown in Figure S1d, the user can manually add an 8- μ L droplet inside the hydrophilic ring to pool the TMT-labeled single-cell samples and transfer it into an autosampler vial for LC injection. Recently, Schoof et al. and Liang et al. have demonstrated the Opentrons OT-2 liquid handler can reliably pipette low- μ L-scale solutions for preparing single-cell samples. Similarly, TMT pooling step for the N2 chip can be automated with conventional LC systems using the OT-2 robot.”

2. Figure 3f: For the SNR increases for the N2 chip vs. Tsai et al and Dou et al – what is the contribution of the new TMT tag structure to this increase in signal? The TMTpro reagents fragment more easily and produce higher reporter ion signal. It’s possible that the increase in SNR is due primarily to the new tag structure and not the advances of the N2 chip. Perhaps the authors could demonstrate that their collision settings for TMTpro generate similar levels of TMT reporter ion signal as compared to the previous settings used for TMT tags.

The reviewer raises a good point here. The contribution of TMT reagents to the SNR enhancement was overlooked in our original manuscript. Indeed, as pointed out by the reviewer and recent study on TMT-based scProteomics (*Hartlmayr et al., Biorxiv, 2021*), the new TMTpro 16plex can generate higher SNRs compared to TMT 10plex.

To understand the contribution of the TMT reagent, we labeled the same lysates with both TMT 10plex and TMTpro 16plex and analyzed them on the same MS (QE plus) using four different normalized HCD collision energy. We compared both, the reporter ion signal intensities and SNRs from fragmentation spectra between the two reagents. As shown in Figure S4 (and below), we consistently observed higher signal intensities and SNRs with TMTpro. These results agreed with the observation by Hartlmayr et al. (*Biorxiv, 2021*). However, we observed the differences were much larger at lower HCD energy levels: the SNRs were increased by 212%, 119%, 67%, and 66% at HCD energies of 26%, 29%, 32%, and 35%, respectively. Because we used similar normalized HCD collision energy (34% and 35%) in our current N2 chip and previous nanowell chip-based work, we reason the TMTpro reagent could lead to a similar improvement of ~66%. Considering that the median SNR increases of 106% and 125% between data from the N2 chip and previous nanowell chip, we conclude the TMTpro reagent accounts for ~40–50% of the total enhancement. We modified text in the result section of the revised manuscript and added a new Figure S4 to clearly distinguish the contribution of the new TMT reagent to the improved metrics.

Figure S4. Comparison of the signal intensities and SNRs between TMT 10plex and TMTpro 16plex-labeled peptides. (a) Experimental designs. (b) The MS 1 signal difference between the two TMT reagents. (c) The comparison of median log₂-transformed report ion intensities and (d) median SNRs at different normalized HCD collision energies.

3. Cell surface markers with scProteomics: The authors demonstrate that they were able to detect membrane proteins that were specific for one cell type as compared to the other two cell types. However, I wonder how specific these protein markers would be compared to various tissues within a mouse. There has been previously catalogued proteomes of various mouse tissues. It would be great to understand if these membrane proteins has been detected specifically in one of those mouse tissues because it would hint at a usable specificity beyond this three cell system.

We thank the Reviewer for this comment. Because most of the proteomes reported for mouse tissues are not cell-type-specific, we attempted to search for these membrane protein markers in human protein atlas (tissue atlas) based on immunohistochemistry staining (<https://www.proteinatlas.org/>). We speculated human and mouse share many similarities in terms of cell types and protein expression patterns and indeed found most of the protein markers for similar cell types in human tissues agreed with our scProteomics data. We observed EZRI and JAM1 were highly enriched in epithelial cells. Immune-cell-related markers CD14, CD68, CYBA (Uniprot protein name: CY24A_Human) were specifically enriched to macrophage cells in the human lung. We have included a new supplementary figure S9 and additional discussion in the result section of the revised manuscript as indicated below.

“We verified the localization of the markers on human immunoperoxidase histology images generated by the Human Protein Atlas focusing on respiratory organs (lung, bronchi, and nasopharynx). While the general organization of the lung differs in human and mice (e.g. number of lobes, airway and bronchi organization), the cell types composing the organ are almost identical as evidenced in a scRNA-seq study. Thus, we speculated the human and mice cells

share many similarities in terms of protein expression patterns. As anticipated, the localization of the protein markers for similar cell types in human tissues is in agreement with our scProteomics data (e.g., C10 and RAW). Both EZRI and JAM1 enriched in C10 are localized in human epithelial cells (Figure S9). The immune-cell-related markers, CD14, CD68, and CYBA (Uniprot protein name: CY24A_Human), are specifically localized in macrophage cells in human lung tissues. Together, these results demonstrated that cell-type-specific surface markers can be effectively identified by combining scProteomics with subcellular-localization information.”

Figure S9. Immunohistochemistry staining images of highly enriched membrane protein makers on specific cell types. Upper images show EZRI and JAM1 enriched in C10 cells are localized in epithelial cells of human nasopharynx tissues; Bottom images show CD14, CD68, and CYBA (Uniprot name: CY24A_Human) enriched in RAW cells specifically localized in macrophase cells of human lung tissue. All the images were downloaded from human protein atlas database (<https://www.proteinatlas.org/>).

Likewise, if there is RNAseq data available for these cells – would these markers have been predicted by the RNAseq data? The real value for scProteomics would be if these markers were not revealed by the RNAseq analysis.

We agree with the Reviewer’s comment here. In the revised manuscript, we re-analyzed publicly available scRNA-seq data on C10 and Raw cells. To assess if mRNA and protein measurements provide the same list of predicted membrane protein markers, we performed a T-test between the mRNA abundance for two cell types and matched the significant proteins to the same subcellular-component database. The overlaps of the proteins between the two measurements were moderate for both cell types. Less than 32.5% of protein targets predicted by RNA-seq were found by scProteomics, indicating mRNA abundances cannot precisely infer membrane protein abundances. Interestingly, both protein and mRNA measurements identified

the six proteins shown in Figure 5 as significantly enriched markers. Overall, our analysis suggests the combination of the two modalities provides the most reliable membrane protein markers. To address the Reviewer's comments, we have added a new section and a new Figure 6 in the revised manuscript as indicated below.

“Finally, we assessed if mRNA and protein measurements predict the same membrane protein markers. After matching to the same subcellular-localization database, scRNA-seq measurements identified 30 membrane proteins for RAW cells and 40 proteins for C10 cells (Supplementary Table 2). The overlap between the two measurements was moderate for both cell types (Figure 6f). Less than 32.5% protein targets predicted by RNA-seq were found by proteomics measurements, indicating mRNA abundances cannot precisely infer membrane protein abundances. Interestingly, both protein and mRNA measurements identified the six proteins shown in Figure 5 as significantly enriched markers. Overall, our analysis suggests the combination of the two modalities provides the most reliable membrane protein markers.”

Reviewer #3

The authors report the nested nanoPOTS (N2) chip for higher throughput and sensitivity over the group's previously reported nanoPOTS platform. The N2 chip demonstrates formidable improvements in sample preparation for single-cell mass spectrometry (e.g., >10x improvement in throughput, 15% improvement in proteome coverage, and ~230% improvement in protein recovery compared to the previous nanoPOTS design) by employing a single cell and nanoliter volume dispensing instrument (cellenONE). This work has the potential to substantially influence the field of single-cell analysis as researchers turn their head toward single-cell proteomics as a rising competitor and complementary technology to single-cell genomics. Moreover, by using commercially and publicly available equipment and software, the authors prime the nanoPOTS technology for broader adoption. Overall, this manuscript provides an excellent introduction to the field of single-cell mass spectrometry for a general scientific audience, is well written and organized, provides sufficient detail for reproducibility, and appropriate statistical analysis. Therefore, this publication is well suited for publication in Nature Communications after the following minor comments are addressed.

Minor Comments:

1. Abstract, page 2, line 22: Define TMT when it first appears.

Corrected as suggested.

2. Figure 1a description, page 5, line 82: “an TMT set” should be “a TMT set”

Corrected as suggested.

3. Methods, page 6, line 97: “incubate” should be “incubated”

Corrected as suggested.

4. Results, page 19, lines 382-383: Make it more clear here that the low sensitivity and reproducibility is referring to data acquired with the previously published nanowell chip and that the N2 chip offers an advantage. Currently, it reads as if the N2 chip has low sensitivity and reproducibility.

We thank the Reviewer for this comment. We have modified the text to read “likely due to low sensitivity and reproducibility of the previous nanowell devices and workflows.”

REVIEWERS' COMMENTS

Reviewer #1 (Remarks to the Author):

In the revised version of their manuscript on single cell proteomics, Woo et al have included additional experimental data, new text paragraphs and clarification of several points. However, the main concern as to show biologically meaningful improvements over their previous platform is still missing. The main selling point is the implementation of the workflow on a commercially available instrument (CellenOne), which could indeed make the technology more accessible to the wider scientific community. But how would newcomers in the field get access to the NanoPOTS chips? Will they be distributed by the authors, by Cellenion SASU or potentially even by a spinoff company? As long as this is not the case, widespread use of the platform established here won't be possible.

To me it still seems as if this manuscript describes intermediate results rather than a finished story. According to the authors, the demonstration of a new biological application (e.g. distinguishing cell types or samples that could NOT be distinguished previously) was not possible due to the COVID-19 crisis. However, at the same time they write "We have initiated several collaborative efforts whereby we are applying our novel scProteomic workflow to address the most challenging biological questions regarding cell typing, toxin response heterogeneity, and cell differentiation". Furthermore, they agree to my previous comment that further improvements of the platform could simply be achieved by reducing the nanowell size to 0.25 mm in diameter. While the authors claim that this would not be a major leap forward, it would still present a 2-fold reduction in diameter, which is the exact same increment as implemented between their 2018 Nature Communications paper and the present manuscript. Taken together, I believe that more impactful data could already be presented. Also note that no improvements in protein coverage have been achieved, which seems to be a parameter that has not been sufficiently addressed.

Minor things:

- I very much appreciate that the authors have now included a comparison of scRNAseq and sc proteomics data (new Fig. 6). However, the fact that the proteomics data shows much less variation between different cells might indicate a missing ability to detect subtle differences based on rather sparse data sets.

- The statement "the adaptive immune system was only detected at transcriptome level" sounds cryptic and requires further clarification. Which exact genes are the authors referring to?

Reviewer #2 (Remarks to the Author):

The authors present an updated and improved manuscript describing the N2 chip as a substrate for single cell proteomic sample preparation. As stated before, the improvements made to the chip as well as the coupling to a commercially available liquid handler are important advancements that will enable broader adoption of this technique. The updates to the manuscript have substantially improved it, particularly the comparison to RNAseq data. Demonstrating that additional information can be learned from single cell proteomics compared to RNAseq is an important step toward the adoption of single cell proteomics more broadly. Additionally, the response to my previous question regarding to the contribution of TMTpro to the improvement in signal been answered in a detailed manner and presents transparency to the source of signal improvements. I do not have any further comments and would support publication in its current form.

Point-by-Point Response to the Reviewer Comments

Reviewer comments in italics, responses in blue.

Reviewer: #1

In the revised version of their manuscript on single cell proteomics, Woo et al have included additional experimental data, new text paragraphs and clarification of several points. However, the main concern as to show biologically meaningful improvements over their previous platform is still missing. The main selling point is the implementation of the workflow on a commercially available instrument (CellenOne), which could indeed make the technology more accessible to the wider scientific community. But how would newcomers in the field get access to the NanoPOTS chips? Will they be distributed by the authors, by Cellenion SASU or potentially even by a spinoff company? As long as this is not the case, widespread use of the platform established here won't be possible.

We thank the Reviewer for raising the important point. We agree the rapid dissemination of newly developed technologies is critical for the science community. This is the main reason we implemented the nested nanoPOTS workflow on a commercially available instrument and closely collaborated with the company to make the workflow to be easily adopted.

We are delighted to share an exciting news that the Germany-based company (Sciencion) and its subsidiary (Cellenion) have signed an exclusive licensing agreement with PNNL. The collaborative agreement aims to prepare nanoPOTS for commercial use (<https://www.cellenion.com/scienion-and-cellenion-exclusively-license-pnnl-developed-tech-and-partner-to-accelerate-sample-processing-for-mass-spectrometry/>). We hope the agreement could accelerate the application of single-cell proteomics in many biology areas.

We want to emphasize that the main innovation (selling point) of the manuscript is the nested nanoPOTS design and new sample preparation workflow, which can significantly improve the throughput, robustness, sensitivity, and reproducibility of single-cell proteomics.

To me it still seems as if this manuscript describes intermediate results rather than a finished story. According to the authors, the demonstration of a new biological application (e.g. distinguishing cell types or samples that could NOT be distinguished previously) was not possible due to the COVID-19 crisis. However, at the same time they write "We have initiated several collaborative efforts whereby we are applying our novel scProteomic workflow to address the most challenging biological questions regarding cell typing, toxin response heterogeneity, and cell differentiation". Furthermore, they agree to my previous comment that further improvements of the platform could simply be achieved by reducing the nanowell size to 0.25 mm in diameter. While the authors claim that this would not be a major leap forward, it would still present a 2-fold reduction in diameter, which is the exact same increment as implemented between their 2018 Nature Communications paper and the present manuscript. Taken together, I believe that more impactful data could already be presented. Also note that no

improvements in protein coverage have been achieved, which seems to be a parameter that has not been sufficiently addressed.

As we addressed in our previous response letter, further reduction in nanowell size would not only lead to improved sample recovery, but also allow more cells to be prepared in single chips. We want to clarify that the reduction of nanowell diameters by twice is a significant improvement, which corresponds to a 75% reduction in surface areas and a 85% reduction of processing volumes. In the current manuscript, we demonstrated that a two-fold reduction from 1 mm to 0.5 mm enabled ~230% improvement in protein recovery, 6-fold increase in throughput, and 15% improvement in proteome coverage.

In the revised manuscript, we discussed more on the further improvement on the nanoPOTS technologies for single cell proteomics as below:

“In the near future, we envision the development of higher capacity N2 chips and/or stable isotope isobaric labeling reagents for hyperplexing scProteomics experiments (e.g., over 1000 cells per chip containing 5×5 array and 40 total clusters). A parallel droplet dispensing system could be developed for ultra-high throughput single cell preparation. To further improve sample recovery and digestion efficiency, the nanowell diameters can be reduced to 0.2 mm or below, which corresponds to total droplet volumes of < 5 nL. New microfluidic strategies could be developed to minimize droplet evaporation and increase droplet dispensing precision. The proteome coverages could be increased with lower-flow LC system, advanced MS instrumentation, and data analysis strategies.”

Minor things::

- *I very much appreciate that the authors have now included a comparison of scRNAseq and sc proteomics data (new Fig. 6). However, the fact that the proteomics data shows much less variation between different cells might indicate a missing ability to detect subtle differences based on rather sparse data sets.*

The low variation in protein abundance is expected, because we employed cells cultured under identical condition. Single cell having a stable proteome is biologically meaningful, as proteins are required at all time to maintain biological process in cells. On the other hand, mRNAs are not required at all time. In fact, many studies indicated that mRNAs are synthesized in short but intense bursts of transcription (Mol. Cell. 2015, 58, 147), which lead to high variations between single cells.

Therefore, in contrary to the Reviewer’s comment, our results demonstrated the present single-cell proteomics workflow is highly reproducible and precise. It can be used to detect subtle difference of cell phenotype when applying to a perturbation study.

- The statement “the adaptive immune system was only detected at transcriptome level” sounds cryptic and requires further clarification. Which exact genes are the authors referring to?

In the revised manuscript, we provided full lists of genes in each enriched Reactome pathway in Source Data 2. As showing below, 17 genes in “adaptive immune system” are significantly enriched (P value of 0.028) in transcriptome level of C10 cells.

		Reactome ID	Term description	Observed gene count	Background gene count	-Log10 FDR	Matching proteins or genes in your network (labels)
Pathway enrichment in C10	N2 proteomics data	MMU-1428517	The citric acid (TCA) cycle and respiratory electron transport	19	121	9.15	Etfb,Cs,Ndufs2,Alp5e,Mdh2,Ndufs6,Atp5j,Glo1,Ndufs1,Ndufs5,Ndufc2,Pdha1,Efta,Suclg1,Ndufa8,Vdac1,Atp5d,Atp5c1,Ndufb5
		MMU-1168372	Downstream signaling events of B Cell Receptor (BCR)	14	59	8.82	Psmb4,Psmb1,Psmc3,Psmc6,Psmc6,Psmc6,Psmc5,Psmc5,Psmc5,Psmc5,Psmc7,Psmc8,Psmc4,Psmc3
		MMU-109581	Apoptosis	14	93	6.73	Bcap31,Dynll1,Ywhab,Dnll,Dynll2,Acin1,Ywhaz,Vim,Gsn,Ywhag,Ywhae,Casp3,Ywhag,H1f0
		MMU-109582	Hemostasis	28	484	5.31	Tubb6,Col1a1,Wdr1,Cav1,Capza2,Crk,Vcl,Ak3,Cyb5r1,Flna,Anxa2,S100a10,F11r,Lamp2,Fn1,Prkar1a,Mapk1,Actn4,Igfb1,Capzb,Hdac1,Cd47,Cd63,Tagln2,Gng12,Sri,Slc3a2,Glg1
		MMU-168249	Innate Immune System	16	246	3.48	Cnn2,Surf4,Rab5c,Rab10,Npc2,Prdx4,S100a11,Ddoat,Pdap1,Lamp1,Pyg,b.Txnnc5,Ckap4,Bst2,Myo1c,Palah1b2
		MMU-199977	ER to Golgi Anterograde Transport	7	69	2.38	Kdelr1,Lman2,Trappc3,Arnc1,Copb2,Lman1,Tmed9
		MMU-381426	Regulation of Insulin-like Growth Factor (IGF) transport and uptake by Insulin-like Growth Factor Binding Proteins (IGFBPs)	5	40	2.01	Rcn1,Calu,Nucb1,Lgals1,Prkcs
		MMU-156827	L13a-mediated translational silencing of Ceruloplasmin expression	5	46	1.82	Eif4h,Rps23,Rps21,Rps28,Eif4b
		MMU-1430728	Metabolism	40	1346	1.69	Cyb5r3,Aldh12,Nme2,Aco2,Adsl,Vapa,Aldh18a1,Got1,Echs1,Hadh,Decr1,Alad,Ugdh,Acads,Asns,Rrm1,Cyb5b,Pck2,Mpst,Gns,Ephx1,Akd,Ugt1a7c,Phgdh,Ak1,Chpf,Oat,Pgm1,Aldh1a1,Marcks,Acadv1,Ass1,Ugp2,Gaa,Cpne1,Aip,Sumo2,Nme1,Hadha,Slc25a12
	scRNA-seq data	MMU-1430728	Metabolism	51	1346	5.31	G6pdx,Por,Slc25a11,Acads,Lta4h,Cyb5r3,Aldh12,Pik,Nt5c,Galk1,Aco2,Hexb,Hsd17b4,Cp11a,Aldh18a1,Gsto1,Got1,Echs1,Rae1,Hadh,Paps1,Apoa1bp,Decr1,Cpne3,Alad,Mecr,Ugdh,Asns,Hlbadh,Pck2,Ephx1,Glb1,Ugt1a7c,Ak1,Ethe1,Chpf,Oat,Aldh6a1,Pgm1,Aldh1a1,Marcks,Ikh1,Hk1,Acadv1,Ugp2,Gaa,Cpne1,Aip,Hadha,Tp11
		MMU-109582	Hemostasis	26	484	4.92	Col1a1,Wdr1,Ppp2r1a,Cav1,Spacc,Actn1,Vcl,Ak3,Cyb5r1,F11r,Mapk3,Lamp2,Fn1,Prkar1a,Mapk1,Actn4,Bsg,Gnas,Igfb1,Ehd2,Cd47,Cd63,Psap,Tagln2,Gng12,Sri
		MMU-199977	ER to Golgi Anterograde Transport	10	69	4.91	Napa,Arf5,Lman2,Arfgap1,Trappc3,Copb2,Copz1,Tmed9,Copa,Mcf2
		MMU-168249	Innate Immune System	17	246	4.57	Cnn2,Cstb,Srp14,Surf4,Npc2,Ostf1,Prdx4,Cat,Erp44,Pyg,b,Val1,Bst2,Pgrmc1,Cap1,Myo1c,Atox1,Faf2
		MMU-1428517	The citric acid (TCA) cycle and respiratory electron transport	11	121	3.89	Ogdh,Etfb,Acad9,Sdha,Aco2,Ndufc2,Pdha1,Efta,Me1,Vdac1,Sucla2
		MMU-1280218	Adaptive Immune System	17	376	2.55	Npepps,Ap1b1,Dync1h1,Sar1b,Lgmn,Ctst,Xdh,Vcam1,Sec13,Actr1a,Dync1i2,Sec31a,Hspa5,Ctsa,Ube2h,Dctn1,Ctsd
		MMU-381426	Regulation of Insulin-like Growth Factor (IGF) transport and uptake by Insulin-like Growth Factor Binding Proteins (IGFBPs)	5	40	2.21	Rcn1,Calu,Nucb1,Lgals1,Prkcs
		MMU-109581	Apoptosis	7	93	2.06	Cttnb1,Dnll,Dynll2,Gsn,Ywhae,Bak1,H1f0
MMU-156827		L13a-mediated translational silencing of Ceruloplasmin expression	15	46	10.22	Pabpc1,Rps11,Rps9,Eif3h,Rps3,Eif3f,Rps4f,Eif3l,Rps7,Rps25,Rps2,Eif3b,Rps10,Rpl13a,Rps20	
Pathway enrichment in RAW	N2 proteomics data	MMU-1430728	Metabolism	65	1346	7.67	Hk2,Ckb,Bvra,G6pdx,Lbr,Adh5,Sm,Aprr,Aco113,Acat2,Pgam1,Lpl,Adss,Slc25a5,Ppa1,Txnrd1,Pik,Nampt,Galk1,Mhfd1,Glxr,Hexb,Dhfr,Glud1,Tkt,Esd,Psat1,Esyt1,Taldo1,Atic,Rab14,Fabp5,Gmps,Paps1,Apoa1bp,Cpne3,Scp2,Mecr,Prps1,Pgls,Pdck,Akr1b8,Nup54,Ptges3,Fasn,Alox5ap,Ethe1,Eno1,Impdh2,Fdps,Pgd,Aldh6a1,Mat2a,Ikh1,Hk1,Itpa,Aldoc,Acly,Pfkp,Tpr,Cndp2,Nup155,Tp11,Ube2i,Adssl1
		MMU-72163	mRNA Splicing - Major Pathway	14	82	6.76	Sf3a1,Hnmpa0,Snrpb2,Prpf31,Prpf8,Cdc5l,Sf3a3,Sf3b3,Hnmpu,Prpf40a,Snrpa,Puf60,Snrnp200,Tra2b
		MMU-1428517	The citric acid (TCA) cycle and respiratory electron transport	12	121	3.59	Acad9,Sdha,Pdhb,Ndufa10,Dist,Ndufa9,Ikh2,Ldha,Did,Ndufv2,Sucla2,Ikh3a
		MMU-6791226	Major pathway of rRNA processing in the nucleolus and cytosol	8	67	2.80	Bop1,Bysl,Wdr75,Ncl,Ebna1bp2,Fbl,Gnl3,Nop56
		MMU-109581	Apoptosis	8	93	2.01	Ywhah,Bax,Hist1h1d,Hist1h1c,Cd14,Hist1h1e,Hmgpb2,Hist1h1b
		MMU-141424	Amplification of signal from the kinetochores	6	60	1.77	Ranbp2,Nudc,Nup160,Rcc2,Xpo1,Rangap1
		MMU-168249	Innate Immune System	13	246	1.74	Ifi2,Ctsz,Cand1,Ostf1,Pa2g4,Cct8,Cat,Pin1,Cct2,Sugt1,Cap1,Anpep,Cd68
		MMU-1222556	ROS, RNS production in phagocytes	4	29	1.59	Atp6v1b2,Cybb,Cyba,Ncf4
		MMU-141405	Inhibition of the proteolytic activity of APC/C required for the onset of anaphase by mitotic spindle checkpoint components	2	5	1.37	Bub3,Mad21
	scRNA-seq data	MMU-156827	L13a-mediated translational silencing of Ceruloplasmin expression	26	46	24.91	Pabpc1,Rps11,Rps9,Rps19,Eif3e,Rps14,Eif3a,Eif3m,Rps3,Rps4x,Rps23,Rps21,Eif3k,Eif2e1,Rps7,Rps25,Eif2s2,Eif3l,Rps16,Rps28,Rps10,Rpl13a,Rps15a,Rps20,Rps13,Eif4b
		MMU-1428517	The citric acid (TCA) cycle and respiratory electron transport	15	121	6.25	Sdhb,Atp5e,Ndufs6,Pdhb,Ndufs5,Ndufs8,Atp5d,Atp5h,Ikh2,Ldha,Atp5c1,Atp5f1,Ndufb5,Atp5j,Ikh3a
		MMU-72163	mRNA Splicing - Major Pathway	10	82	4.05	Cdc5l,Dhx15,Pcbp1,Prpf40a,Sf3b4,Hnmpa3,Hnmp21,Hnmpa2b1,Tra2b,Hnmp1
		MMU-1168372	Downstream signaling events of B Cell Receptor (BCR)	8	59	3.54	Psmc11,Psmc5,Psmc2,Psmc1,Psmc8,Psmc5,Psmc3,Psmc3
		MMU-1222556	ROS, RNS production in phagocytes	5	29	2.55	Atp6v1f1,Cybb,Cyba,Atp6v1g1,Ncf4
		MMU-109581	Apoptosis	8	93	2.46	Dynll1,Ywhah,Acin1,Ywhaz,Ywhag,Cd14,Hmgb2,Ywhaq
		MMU-1169408	ISG15 antiviral mechanism	3	14	1.73	Eif4e,Eif4g1,Ube2n
		MMU-429914	Deadenylation-dependent mRNA decay	4	30	1.73	Lsm2,Lsm3,Lsm5,Ddx6
MMU-109582		Hemostasis	18	484	1.66	Tubb6,Rhoa,Atp2b1,Sod1,Kif5b,Cd9,Rac2,S100a10,Mif,Cdc42,Fam49b,Pta2g4a,Tuba1b,Aldoa,Hdac1,Ptpn6,Glg1,Cd36	
MMU-2990846	SUMOylation	4	34	1.66	Park7,Hnmpk,Hnmpc,Nop58		
MMU-1430728	Metabolism	38	1346	1.57	Ckb,Lbr,Sm,Aprr,Pgam1,Adss,Slc25a5,Txnrd1,Nme2,Glxr,Dhfr,Glud1,Esd,Taldo1,Fabp5,Scp2,Ak2,Paics,Got2,Pgls,Cyb5b,Akr1b8,Ptges3,Alox5ap,Acot7,Eno1,Impdh2,Fdps,Gpx1,Pgd,Mat2a,Aldoc,Hadhb,Trm112,Sumo2,Tpr,Ube2i,Adssl1		
MMU-6791226	Major pathway of rRNA processing in the nucleolus and cytosol	5	67	1.47	Wdr75,Wdr12,Ncl,Fbl,Gnl3		
MMU-390466	Chaperonin-mediated protein folding	2	8	1.37	Cct3,Cct7